# The cooperative binding of TDP-43 to GU-rich RNA repeats antagonizes TDP-43 aggregation

**Juan Carlos Rengifo-Gonzalez[1], Krystel El Hage[1], Marie-Jeanne Clément[1], Emilie Steiner[1], Vandana Joshi[1], Pierrick Craveur[2], Dominique Durand[3], David Pastré[1]\*, Ahmed Bouhss[1]\*†**

[1]Université Paris-Saclay, INSERM U1204, Univ Evry, Structure-Activité des Biomolécules Normales et Pathologiques (SABNP), Evry-Courcouronnes, France; [2]SYNSIGHT, Evry-Courcouronnes, France; [3]Université Paris-Saclay, CEA, CNRS, Institute for Integrative Biology of the Cell (I2BC), Gif-sur-Yvette, France

**Abstract** TDP-43 is a nuclear RNA-binding protein that forms neuronal cytoplasmic inclusions in two major neurodegenerative diseases, ALS and FTLD. While the self-assembly of TDP-43 by its structured N-terminal and intrinsically disordered C-terminal domains has been widely studied, the mechanism by which mRNA preserves TDP-43 solubility in the nucleus has not been addressed. Here, we demonstrate that tandem RNA recognition motifs of TDP-43 bind to long GU-repeats in a cooperative manner through intermolecular interactions. Moreover, using mutants whose cooperativity is impaired, we found that the cooperative binding of TDP-43 to mRNA may be critical to maintain the solubility of TDP-43 in the nucleus and the miscibility of TDP-43 in cytoplasmic stress granules. We anticipate that the knowledge of a higher order assembly of TDP-43 on mRNA may clarify its role in intron processing and provide a means of interfering with the cytoplasmic aggregation of TDP-43.

**\*For correspondence:**
david.pastre@univ-evry.fr (DP);
ahmed.bouhss@univ-evry.fr (AB)

†Lead contact (Ahmed Bouhss)

**Competing interest:** The authors declare that no competing interests exist.

## Introduction

In comparison to other proteins, many RNA-binding proteins (RBP) harbor low-complexity domains (LCD) that initiate weak multivalent interactions leading to the assembly of liquid-like membraneless organelles, notably in the mammalian cell nucleus (*Hyman et al., 2014*; *Lin et al., 2015*; *Boeynaems et al., 2018*; *Xue et al., 2019*). Membraneless organelles induced by RBPs are involved in critical cellular functions such as the biogenesis and processing of mRNAs in the nucleus (*Gerstberger et al., 2014*; *Swain et al., 2016*). Since TAR DNA-binding protein 43 (TDP-43), a nuclear mRNA-binding protein with a self-attracting LCD (*Cao et al., 2019*; *Tollervey et al., 2011*), was associated to amyotrophic lateral sclerosis (ALS) and frontotemporal lobar degeneration (FTLD) pathologies (*Neumann et al., 2006*; *Arai et al., 2006*; *Sreedharan et al., 2008*; *Vogler et al., 2018*; *Chou et al., 2018*), the notion that deregulated RBP assemblies may be responsible for RBP aggregation has made its way (*Li et al., 2013*; *Patel et al., 2015*). In support to this, the majority of pathological mutations associated with TDP-43 are located in its self-adhesive C-terminal LCD (*Van Deerlin et al., 2008*). Many structural analyses focused on pathological TDP-43 mutations have also indicated a critical role of the TDP-43 LCD in the aggregation process (*Cao et al., 2019*; *Conicella et al., 2016*; *Guenther et al., 2018*; *Conicella et al., 2020*; *Watanabe et al., 2020*).

However, despite the link between the TDP-43 LCD and several neurodegenerative diseases, the precise mechanism leading to the formation of cytoplasmic aggregates of TDP-43 remains obscure. The known mutations located in the TDP-43 LCD, associated with neurodegenerative diseases, still

allow normal cognitive functions in adults before aging (*Li et al., 2013*; *Patel et al., 2015*). Therefore, other mutations leading to a more severe phenotype may be useful to probe the structural basis leading to aggregation in cells and to identify possible means to interfere with a putative gain and (or) loss of functions of TDP-43 inclusions in neurons. In recent in vitro studies, the dimerization of the structured N-terminal was proposed to promote a head-to-tail aggregation of TDP-43, together with the self-adhesive LCD (*Wang et al., 2018a*; *Afroz et al., 2017*; *Loughlin and Wilce, 2019*). Here, we consider the role of mRNA which is the principal partner of TDP-43 in cells (*Polymenidou et al., 2011*; *Lukavsky et al., 2013*; *Schmidt et al., 2019*). Under physiological conditions, TDP-43 is mostly associated with nuclear RNA, considered as a negative regulator of TDP-43 condensates (*Maharana et al., 2018*). A high ratio of RNA/protein in the nucleus promotes dynamic and reversible higher order assemblies of RBPs such as TDP-43, while lower nuclear RNA levels causes excessive phase separation and the formation of cytotoxic aggregates (*Maharana et al., 2018*). In the nucleus, TDP-43 binds to GU-rich sequences in introns (*Tollervey et al., 2011*; *Polymenidou et al., 2011*), with a high affinity through its tandem RRM domains (*Lukavsky et al., 2013*). Notably, transcripts with long introns display multiple GU-rich binding sites (*Polymenidou et al., 2011*). TDP-43 multimerization on long GU-rich repeats may ensure the proper packaging of introns to facilitate their processing by the spliceosome (*Ishiguro et al., 2017*). In the cytoplasm, dynamic liquid-like mRNA-rich compartments (*Buchan and Parker, 2009*), called stress granules, appear after a wide variety of stress (*Colombrita et al., 2009*; *Dewey et al., 2011*) including oxidative stress and hypoxia which may occur during aging in neurons. As proposed in recent studies, the recruitment of TDP-43 in mRNA-rich stress granules may preserve the solubility of TDP-43 in the cytoplasm (*Mann et al., 2019*; *Zacco et al., 2019*), like nuclear RNA under physiological conditions (*Maharana et al., 2018*; *Wang et al., 2020*). Consistent with the notion that mRNA keeps TDP-43 in a soluble state, the specific binding of TDP-43 RRM1–2 to GU-rich RNA limits TDP-43 aggregation in vitro (*French et al., 2019*). In addition, a pathological mutation, K181E, located in the linker between RRM1 and RRM2 domains, leads to TDP-43 aggregation and to a reduced affinity for its RNA targets (*Chen et al., 2019*).

In other models, the recruitment of TDP-43 in stress granules rather plays a negative role in neurodegenerative diseases by acting as crucibles in which TDP-43 is concentrated (*Li et al., 2013*; *Zhang et al., 2015*; *Langdon and Qiu, 2018*; *Van Treeck et al., 2018*; *Dobra et al., 2018*; *François-Moutal et al., 2019*). However, since stress granules are dynamic and reversible compartments, the genesis of insoluble TDP-43 inclusions may as well take place outside (*Gasset-Rosa et al., 2019*) or inside stress granules (*Fang et al., 2019*; *McGurk et al., 2018*).

Here, we devised that an unaltered binding of TDP-43 to mRNA retains TDP-43 in a soluble state to prevent its aggregation whatever in the cytoplasm or in the nucleus. We then wondered whether a specific structural organization of TDP-43 in association with mRNA may contribute to preserve TDP-43 solubility under physiological conditions. To explore this idea, we dissected the binding of multiple TDP-43 to a long RNA and then discovered the cooperative binding of the tandem TDP-43 RNA Recognition Motifs (RRM1–2), to long GU repeats. A cooperative binding to RNA is not unique to TDP-43 but is also found in many other RBPs. Indeed, most RBPs have a low specificity for short sequences of few nucleotides (*Singh and Valcárcel, 2005*). A cooperative association of RBPs to RNA may enable their specific recognition of longer RNA sequences, for example, as evidenced through the cooperative association of Unr and Sxl RBPs to the 3'UTR of *msl2* mRNA (*Hennig et al., 2014*). In addition, cooperativity may secure the attachment of RBPs on their mRNA targets. Here, through a structural analysis by NMR spectroscopy of the intermolecular interface between two RRM1–2 monomers, we identified the residues driving the cooperative binding of TDP-43 on long GU-rich sequences. We then probed whether TDP-43 mutants with an impaired cooperative binding to mRNA are imperfectly miscible with wild type TDP-43 in a cellular context. To this end, we analyzed the mixing/demixing between wild-type TDP-43 and selected mutants in cells, using microtubules as nano-platforms (*Maucuer et al., 2018*). Our integrative approach reveals an intermolecular interaction between the loop 3 of RRM1 and a pocket centered around the V220 in RRM2 thus providing the missing link responsible for the cooperative binding of TDP-43 to mRNA. We also showed that the disruption of the cooperative binding of TDP-43 promotes the assembly of mRNA-poor TDP-43 aggregates in the nucleus and antagonizes the presence of TDP-43 in mRNA stress granules in the cytoplasm. In light of the results presented here, we propose a mechanistic model in which TDP-43, through its cooperative binding to mRNA, prevents the structured N-terminal and C-terminal LCD

domains from inducing TDP-43 aggregation. Unravelling how TDP-43 preserves its solubility to enable normal processing of nuclear mRNA could also provide means to interfere with the pathologic transition leading to protein-pure amyloids or to correct splicing defects in neurons of patients affected by TDP43-positive neurodegenerative diseases.

## Results

## RRM1–2 of TDP-43 binds to long nucleic acid targets in a cooperative manner and forms protein multimers

To understand through which molecular mechanisms TDP-43 targets pyrimidine-rich introns in cells as revealed by CLIP experiments, we considered whether a cooperative association of TDP-43 may take place in long GU-rich repeats to secure the binding of several TDP-43 proteins. The structure of TDP-43 is generally represented with three distinct functional domains: a structured N-terminal domain (NTD), two central RRMs, and a long unstructured C-terminal (LCD) (*Wang et al., 2018b*; *Mompeán et al., 2016*; *Jiang et al., 2017*; *Kuo et al., 2009*, *Figure 1a*). Since the two well-conserved RRM domains, RRM-1 and -2 (RRM1–2), bind to GU-rich RNA sequences with a high affinity (*Lukavsky et al., 2013*), we focused our attention on RRM1–2. However, so far, only structural data about the binding of a single RRM1–2 to short eight nt-long RNA sequences are available (*Lukavsky et al., 2013*). We then used cross-linking and gel mobility shift assays combined to ITC measurements to document the binding of multiple TDP-43 to long GU-rich repeats.

The presence of higher order assemblies of TDP-43 was first evidenced by cross-linking assays in the presence of a loop containing 24 GT repeats that mimics long GU-rich mRNA sequences functionally targeted by TDP-43 (*Tollervey et al., 2011*; *Humphrey et al., 2017*) and can accommodate the binding of up to four RRM1–2 protein fragments (Q101-G277, *Figure 1a,b*). DNase treatment did not disrupt the complexes preformed in the presence of GT repeats (*Figure 1—figure supplement 1*). Therefore, a direct protein–protein interaction may probably take place in the presence of 24 GT repeats.

To further characterize the higher order assembly of TDP-43 bound to 24 GT repeats, gel mobility shift assays were performed with RRM1–2 in the presence of 24 GT repeats (*Figure 1c*). In addition, isolated RRM1 (Q101-K192) and RRM2 (K176-G277) were prepared to compare their interactions with GT repeats with those of tandem RRM1–2 (*Figure 1a*). RRM1 or RRM2 fragments bind to GT repeats progressively and in an uncooperative manner. However, RRM1–2 forms, abruptly and discretely, successive multimeric intermediates with increasing protein concentrations, a hallmark of a cooperative association. As RRM1 has a high affinity for GT repeats and can also form multimers, RRM1 may drive, by itself, the binding of RRM1–2 to GT repeats through discrete steps. However, RRM1 alone, in contrast to RRM1–2, does not show a discrete binding to 24 GT repeats, suggesting a significant role of RRM2 domain to generate stable RRM1–2 multimers.

Whereas electrophoretic mobility assays and cross-linking analyses reveal the formation of TDP-43 multimers on 24 GT repeats, an assessment of its cooperative binding is missing. Therefore, we applied isothermal titration calorimetry (ITC) (*Figure 1d*, *Supplementary file 3*, *Figure 1—figure supplement 2*). The ITC titration curves of RRM1–2 with 12 GT repeats display two distinct plateaus reflecting the binding of one and two proteins, while only one plateau is observed with 6 GT repeats indicating the binding of only one RRM1–2. In addition, the enthalpy difference ($\Delta H$) resulting from the binding of two RRM1–2 to 12 GT repeats (~ −80 kcal/mol) is more than twice the $\Delta H$ value measured when only one RRM1–2 binds to 6 GT repeats (~ −33 kcal/mol), which most probably indicates an energy benefit due to a cooperative binding. In comparison to RRM1–2, RRM1 or RRM2 fragments can possibly form dimers with 6 GT repeats and at least tetramers with 12 GT repeats but no clear plateaus were observed in ITC curves to reveal the presence of stable multimers. To score the cooperative binding of TDP-43 to GT repeats, we measured the ratio $Kd_1/Kd_2$. $Kd_1$ and $Kd_2$ represent the dissociation constants of the first and second proteins interacting respectively with GT repeats, as measured by ITC (*Brown, 2009*). Since $Kd_2$ is significantly lower than $Kd_1$ for RRM1–2 ($Kd_1/Kd_2 > 100$, *Figure 1d*), RRM1–2 binds to 12 GT repeats with a higher affinity when another RRM1–2 monomer is already associated to GT repeats, which characterizes a marked cooperativity. Interestingly, the ratio $Kd_1/Kd_2$ was not significantly higher for isolated RRMs, RRM1 and RRM2 fragments ($Kd_1/Kd_2 = 8.9$ and 0.05, respectively) than for RRM1–2 ($Kd_1/Kd_2 = 128$).

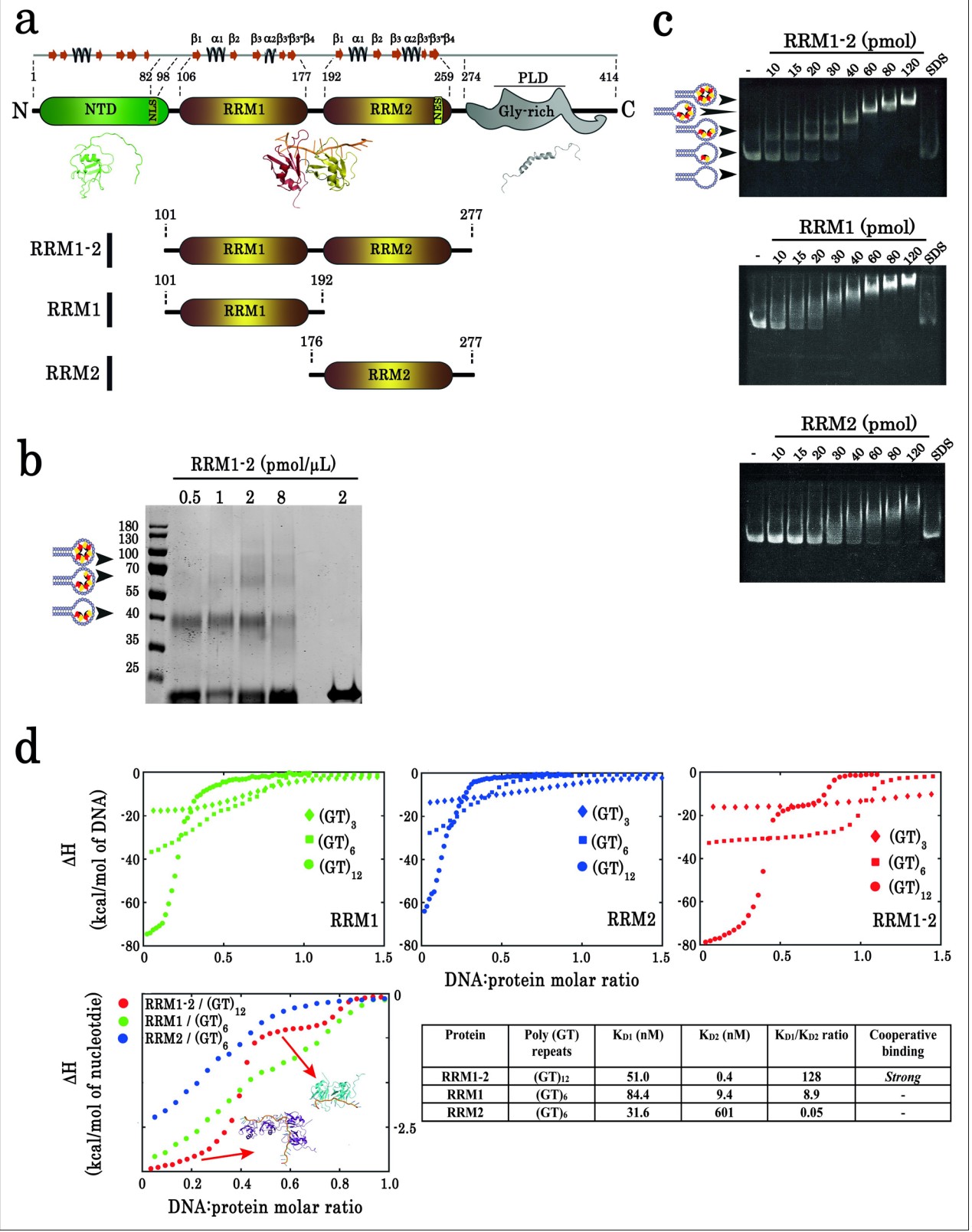

**Figure 1.** Biochemical characterization of TDP-43 fragments bound to oligonucleotide targets. (**a**) Schematic representation of TDP-43 domains. Numbers indicate the boundaries according to the full-length protein sequence (NP_031401). The available 3D structure of N-terminal (PDB 2N4P), RRMs (PDB 4BS2), and C-terminal (PDB 2N3X) are also shown together with β-strands and α-helices. The boundaries of all three recombinant RRM fragments (RRM1–2, RRM1, and RRM2) used in this study are indicated. (**b**) Cross-linking experiments using increasing RRM1–2 protein concentrations,

*Figure 1 continued*

(GT)$_{24}$-loop (10 pmol) and BS3 as cross-linking reagent. Cross-linked proteins are indicated with head-arrows. Last lane corresponds to the experiment in absence of BS3. (**c**) Electrophoretic Mobility-Shift Assay (EMSA) experiments were performed by using increasing protein concentrations and 10 pmol of a stem-loop DNA containing a (GT)$_{24}$ repeats ((GT)$_{24}$-loop). When indicated, the sample was treated with SDS in order to disassemble DNA-protein complexes. Free or protein-containing (GT)$_{24}$-loop are indicated by head-arrows. (**d**) Binding of TDP-43 fragments to (GT)-rich oligonucleotides containing three (diamonds), six (squares) or twelve (circles) GT-repeats was monitored by ITC. At the bottom, plot of ITC titration curves for oligonucleotides that can bind two protein monomers (RRM1/(GT)$_6$, RRM2/(GT)$_6$ and RRM1–2/(GT)$_{12}$). Plateaus corresponding to dimer and monomer states are indicated (red, green and blue curves correspond to RRM1–2, RRM1, and RRM2, respectively). Lower right panel: Kd$_1$/Kd$_2$ ratios obtained from (GT)$_{12}$ or (GT)$_6$ titration data. The binding of RRM1–2 to GT repeats is highly cooperative which is not observed for isolated RRM1 and RRM2. Thermodynamic parameters and ITC statistics are shown in ***Supplementary file 3***. Raw thermograms are shown in ***Figure 1—figure supplement 2***.

The online version of this article includes the following source data and figure supplement(s) for figure 1:

**Source data 1.** SDS-polyacrylamide gel electrophoresis of samples from cross-linking experiments performed in absence of benzonase (see legend of ***Figure 1b***).

**Source data 2.** Electrophoretic Mobility-Shift Assay (EMSA) experiments on RRM1–2 (see legend of ***Figure 1c***).

**Source data 3.** EMSA experiments on RRM1 (see legend of ***Figure 1c***).

**Source data 4.** EMSA experiments on RRM2 (see legend of ***Figure 1c***).

**Source data 5.** ITC data obtained from the binding of TDP-43 fragments to (GT)-rich oligonucleotides (See legend ***Figure 1d***).

**Figure supplement 1.** SDS-polyacrylamide gel electrophoresis of purified proteins and cross-linking experiments.

**Figure supplement 1—source data 1.** SDS-polyacrylamide gel electrophoresis of purified proteins (See legend of ***Figure 1—figure supplement 1a***).

**Figure supplement 1—source data 2.** SDS-polyacrylamide gel electrophoresis of samples from cross-linking experiments performed in presence of benzonase (see legend of ***Figure 1—figure supplement 1b***).

**Figure supplement 2.** Raw calorigrams corresponding to the ITC curves displayed in ***Figure 1d***.

**Figure supplement 2—source data 1.** ITC data obtained from the binding of TDP-43 fragments to (GT)-rich oligonucleotides (See legend of ***Figure 1—figure supplement 2***).

Together, biochemical and ITC data provide compelling evidence of a cooperative binding of TDP-43 RRM1–2 to GT repeats.

## NMR spectroscopy reveals residues possibly involved in the cooperative binding of TDP-43 to RNA

To provide insights into the structural mechanism leading to the cooperative association of RRM1–2 to GU or GT repeats, we performed a comparative analysis by NMR spectroscopy of RRM1, RRM2, and RRM1–2 residues in interaction with GU sequences of different length (6, 12, and 24 GT repeats). In agreement with the results from ITC experiments, a single RRM1–2 binds to 6 GU repeats mostly through conserved RNA-binding residues (***Lukavsky et al., 2013***; ***Figure 2a***). Twelve GU repeats induce additional chemical shift perturbations (CSPs) and peak broadenings, consistent with the binding of two RRM1–2 (***Figure 2b***). These effects were specific to RRM1–2 as compared to RRM1 or RRM2 alone (***Figure 2c***). Given the presence of CSPs for several RRM1–2 residues located in RRM1 and RRM2 domains, the possibility that RRM1 would by itself generate the TDP-43 multimerization can again be ruled out. Moreover, CSPs revealed that interactions with nucleic acids through conserved RNA-binding residues are preserved when RRM1–2 interacts with 12 GU repeats. Thus, both RRM1 and RRM2 bind to RNA, despite the lower affinity of RRM2 than RRM1 for RNA (***Figure 2c***).

To identify TDP-43 RRM1–2 residues involved in the cooperative binding of TDP-43, we selected residues displaying significant CSPs in the presence of 12 GU repeats compared to 6 GU repeats which can accept only one RRM-1–2 (***Figure 2b*** and ***Figure 2—figure supplement 2***). Another criterion used for this selection was the presence of additional CSPs and peak broadening variations when the number of GU repeats was increased from 12 to 24, which may reflect the binding of more than two RRM1–2 to 24 GU repeats (***Figure 2—figure supplement 1***). In addition, according to ITC experiments (***Figure 1d***), CSPs related to a cooperative association should not take place when RRM1 or RRM2 alone interacts with 12 GU repeats instead of RRM1–2. We therefore discarded RRM-1 or RRM-2 residues displaying similar shifts than RRM1–2 residues in the presence of 6 or 12 GU repeats. Finally, conserved RNA-binding residues known to interact directly with RNA were not considered for mutagenesis (***Lukavsky et al., 2013***).

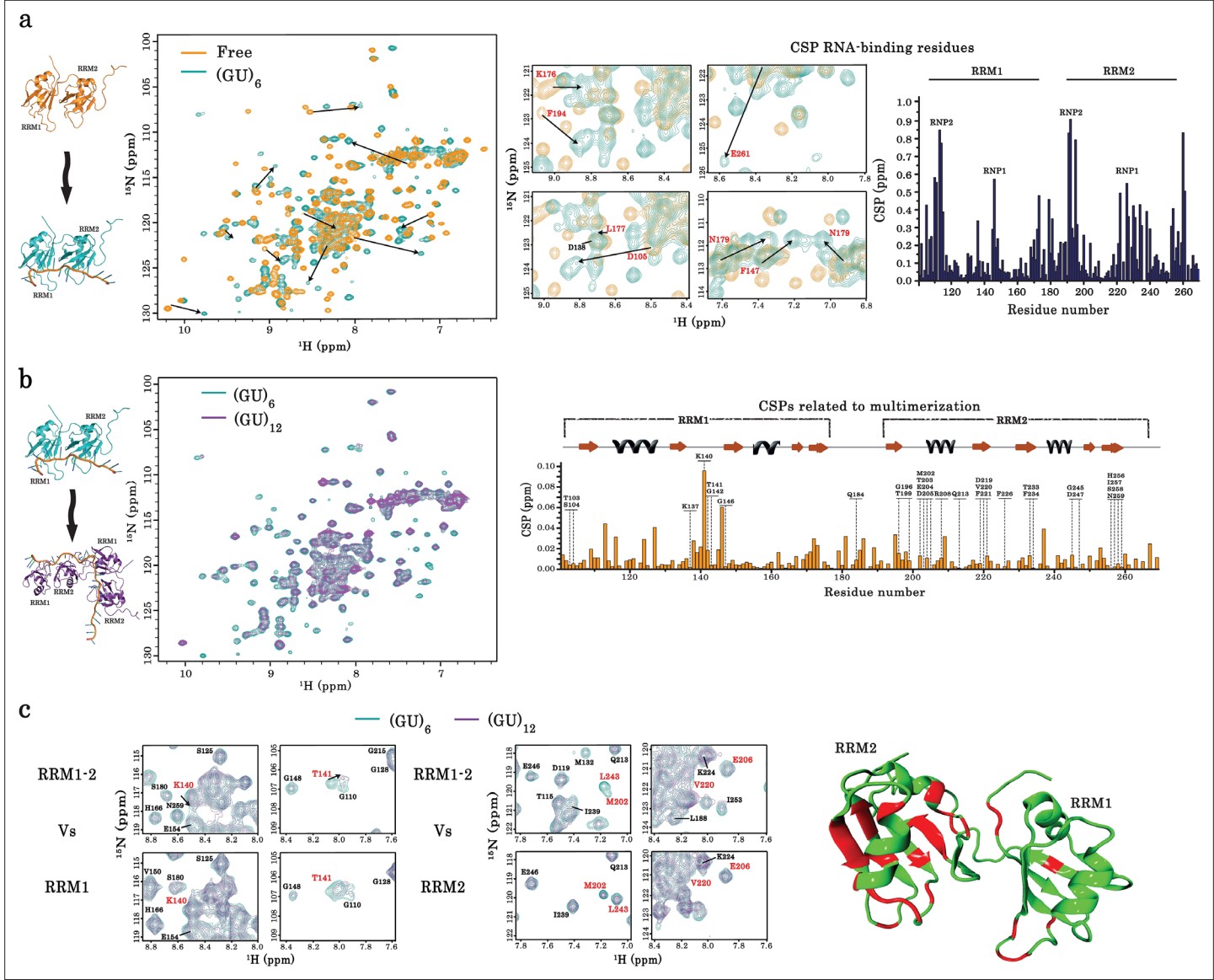

**Figure 2.** Identification of TDP-43 residues involved in its dimerization on GU-repeats. (**a**) NMR spectra of free and bound RRM1–2. *Left*, superimposition of $^1$H-$^{15}$N SOFAST-HMQC spectra of $^{15}$N-labeled RRM1–2 in the free (orange) and (GU)$_6$ RNA-bound (turquoise) forms. Residues displaying the largest chemical shift perturbations (CSP) are indicated by arrows. *Middle*, zoom in on NMR spectra (left) showing the CSPs for some residues (highlighted in red) upon (GU)$_6$ RNA binding. *Right*, plot of CSPs occurring in RRM1–2 upon (GU)$_6$ RNA binding. The combined CSPs were calculated as reported (*Williamson, 2013*) and follow the same trajectories as previously published (*Lukavsky et al., 2013*) for RRM1–2 bound to AUG12 (PDB 4BS2). (**b**) NMR spectra of monomeric and dimeric forms of RRM1–2 bound to GU-repeats. *Left*, superimposition of $^1$H-$^{15}$N SOFAST-HMQC spectra of $^{15}$N-labeled RRM1–2 bound to (GU)$_6$ (turquoise) or (GU)$_{12}$ (magenta). *Right*, combined CSPs, observed for monomeric and dimeric couples, plotted and linked to the secondary structures on top. (**c**) *Left*, zoom in on NMR spectra (**b**) showing RRM1–2 residues displaying particular CSPs, resonance disappearing, or peak broadening (in red) as compared to respective residues in RRM1 or RRM2 fragments. *Right*, all affected residues upon RRM1–2 dimerization are highlighted in red using molecular modelling approaches on RRM1–2 free fragment (see methods). Based on the above comparative NMR study, 28 residues were selected as candidates for mutagenesis approach combined to a detailed cellular and biochemical investigation.

The online version of this article includes the following source data and figure supplement(s) for figure 2:

**Source data 1.** NMR data of RRM1-2 in free and bound forms (See legend of *Figure 2a,b*).

**Figure supplement 1.** Interaction of RRM1–2 protein fragment with 24 GU-repeats.

**Figure supplement 2.** Plot showing the intensity ratios calculated between the $^{15}$N-labeled RRM1–2 bound to (GU)$_6$ or (GU)$_{12}$ RNA.

**Figure supplement 2—source data 1.** NMR data displaying Peak Intensity of RRM1-2 residues in presence of (GU)$_6$ or (GU)$_{12}$ (See legend of *Figure 2— figure supplement 2*).

According to the above-mentioned selection rules, 28 residues likely critical for the homotypic interactions leading to the formation of RRM1–2 multimers in the presence of GU repeats were selected and represented in the RRM1–2 primary structure (*Figure 3b*).

## The 'microtubule bench' reveals residues located in RRM1–2 that play key roles in TDP-43 compartmentalization in HeLa cells

Even if CSPs may be useful to identify critical residues and delineate an interface between two RRM1–2 monomers in the presence of GU repeats, the active contribution of the selected residues to TDP-43 cooperativity remains to be validated. Indeed, unstructured residues may be particularly sensitive to environmental changes following RRM1–2 multimerization without being directly involved in any interaction. Other methods have therefore to be used to decipher which of the identified residues actively participate in the cooperative binding phenomenon of RRM1–2 to RNA, notably in a cellular context.

To probe the putative functional role of the selected residues, after our NMR analysis (*Figure 3b*), in the cooperative binding of TDP-43, we took advantage of a method that we recently developed, namely the microtubule bench (MT bench). MT bench makes use of microtubules as intracellular platforms (*Maucuer et al., 2018*; *Boca et al., 2015*) to analyze the spatial distribution of two different RBPs along microtubules and quantify their mixing/demixing. Given our working hypothesis, disrupting the cooperative binding of RRM1–2 to RNA may lead to the demixing between wild type and mutant of TDP-43. To use this method, wild-type and mutant TDP-43 proteins fused to a microtubule-binding domain are co-expressed in cells (*Figure 3a*). The two TDP-43 proteins are then confined on the microtubule network in cells to probe their interplay. Of note, the confinement of wild-type TDP-43 on microtubules in HeLa cells was already analyzed and led to the formation of reversible mRNA-rich compartments (*Maucuer et al., 2018*). In addition, we noticed the wetting of microtubules with other liquid-like mRNA-rich compartments in the cytoplasm, i.e. stress granules, when wild-type TDP-43 was confined on microtubules (*Maucuer et al., 2018*).

To analyze the role of the 28 TDP-43 residues selected from our structural analysis, the mixing of 34 TDP-43 mutants with wild-type TDP-43 was probed with the MT bench assay (*Figures 2c and 3*, *Figure 3—figure supplement 1*). As a positive control, a mutation in the TDP-43 N-terminal domain involved in TDP-43 self-association (E17A) leads to a significant demixing of E17A mutant with wild-type TDP-43, which emphasizes the sensitivity of the method to detect an impaired TDP-43 multimerization (*Figure 3—figure supplement 1*). In RRM domains, four mutations in the RRM1 sequence, T103A/S104A, K140A/T141A, T141A/G142A, and G146A, and three mutations in the RRM2 sequence, M202A, D205A, and V220A, lead to a remarkable demixing between wild-type and mutant of TDP-43. The other TDP-43 mutants do not significantly change the mixing of TDP-43 mutants with wild-type TDP-43, attesting for the stringency of our method (*Figure 3c*). Interestingly, the active mutations leading to a significant demixing are mostly located away from the intramolecular interface between RRM1 and 2 (*Figure 3d*). Indeed, residues K140, T141, G142, and G146 are located in the long RRM1 loop 3, indicating its possible role in the liquid-liquid phase separation orchestrated by TDP-43.

In summary, the MT bench assay identified nine residues out of the 28 residues selected by NMR which impact significantly the mixing of TDP-43 with itself in cells (*Figure 3c,d*). However, as M202, D205, and V220 are located quite far apart from each other, a possible interacting surface in RRM2 cannot be delineated (the Cα-Cα separation distance is 7.7, 8.8, and 11.9 Å for [D205; V220], [M202; D205], and [M202; V220], respectively; and their side chains are oriented in opposite directions). We therefore sought additional data to document this point.

## The intermolecular interface between RRM1–2 involves an interaction between RRM1 loop 3 and an RRM2 pocket around V220

With the aim to delineate an intermolecular interface, we first decided to use TDP-43 mutants that would be defective in their cooperative association to mRNA in gels shift and ITC assays. A comparison between wild-type and mutant RRM1–2 could then help to identify an interface by using a comparative NMR analysis. We focused our attention on the RRM1 loop 3, as many residues located in this region are critical for the mixing of TDP-43 mutants with wild-type form in cells. The two mutants, T141A/G142A and G146, were selected based on the strongest effect on their demixing with wild-type TDP-43 in cells and the Q213A mutant was added as a negative control. After their production and purification, all the three mutant proteins were soluble (*Figure 1—figure supplement 1*). The

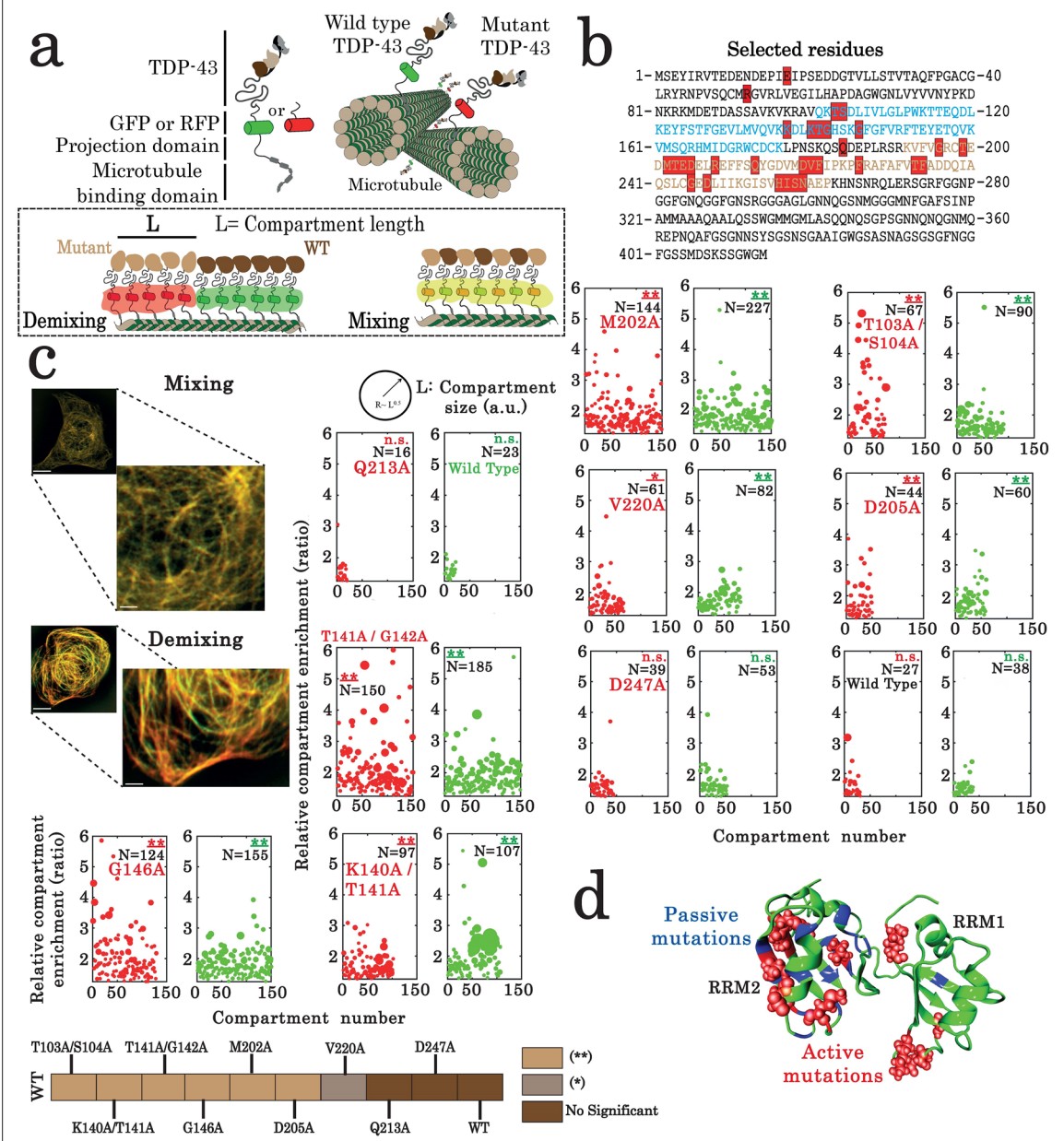

**Figure 3.** Assessment of interactions between wild-type and mutant TDP-43 by using the microtubule bench assay (*Maucuer et al., 2018*). (**a**) Scheme representing the method used to probe homotypic interactions between wild-type and mutant forms of full-length TDP-43. To track their subcellular protein localization, wild-type and mutant TDP-43 were labeled with the Green Fluorescent Protein (GFP) and the Red Fluorescent Protein (RFP), respectively (**b**) Amino acid sequence of full-length TDP-43 showing RRM1 (blue) and RRM2 (brown) motifs. Amino acids subjected to mutagenesis are highlighted by red boxes. (**c**) HeLa cells co-expressing GFP- and RFP-labeled TDP-43 in order to assess mixing/demixing on microtubules. Relative enrichment of both wild-type and mutated TDP-43 compartments is expressed as a function of compartment number according to the procedure described in the Materials and methods section. Several mutations such as G146A display an elevated enrichment and many more compartments than when two wild-type TDP-43 are interacting. Scale bar: 10 μm. p<0.05*; p<0.01** (paired two-sample Kolmogorov–Smirnov test compared to control). (n.s.) non-significant. (N), number of compartments. (**d**) TDP-43 mutants displaying mixing or demixing are referred as 'passive mutations' or 'active mutations', respectively. Demixing denotes a perturbation in interactions between wild-type and mutant of TDP-43. The amino acid residues corresponding to the 'passive mutations' or 'active mutations' are shown in blue or red, respectively.

The online version of this article includes the following source data and figure supplement(s) for figure 3:

**Source data 1.** Assessment of interactions between wild-type and mutant full-length TDP-43 by using the microtubule bench assay (See of legend *Figure 3c*).

**Figure supplement 1.** Assessment of interactions between wild type and mutants of full-length TDP-43 by using the microtubule bench assay (*Maucuer*

*Figure 3 continued on next page*

*Figure 3 continued*

*et al., 2018*).

**Figure supplement 1—source data 1.** Assessment of interactions between wild type and mutants of full-length TDP-43 by using the microtubule bench assay (See legend of *Figure 3—figure supplement 1*).

double mutant in RRM1 loop 3, T141A/G142A, strongly affects the binding of TDP-43 to 24 GT repeats as compared to wild-type and Q213A RRM1–2 proteins (*Figure 4a*). RRM1–2/DNA complexes arise at similar RRM1–2 concentrations for G146A and wild-type RRM1–2, thus indicating that G146A has a similar affinity for GT repeats than wild-type RRM1–2. However, G146A binds to 24 GT repeats with a reduced cooperativity compared to both wild-type RRM1–2 and Q213A mutant. For instance, below saturating concentrations of RRM1–2, we noticed an increase occurrence of lower molecular weight complexes in the case of G146A compared to wild-type RRM1–2 and Q213A (see for example at 20 pmol, *Figure 4a*).

We then analyzed the NMR spectra of T141A/G142A and G146A and compared them to the wild-type RRM1–2 in order to clarify whether mutations, particularly for the double mutant, would not affect its binding property to RNA. The results indicate that most residues in these mutants display changes in their chemical shifts comparable to the wild-type form in the presence of six GU repeats (*Figure 4b*). We then hypothesized that binding differences of the T141A/G142A and the G146A mutants versus wild-type protein detected by gel mobility shift assays (*Figure 4a*) were related to the higher order assembly of RRM1–2 when bound to 12 GT repeats. Consistently, as deduced from the ITC titration curves, T141A/G142A and G146A, when interacting with 12 GT repeats, do not display the typical ITC thermogram profile linked to a stable dimer as exhibited in the case of Q213A or wild-type forms (*Figure 4c*, *Supplementary file 4*, *Figure 4—figure supplement 1* and *Figure 1d*), thus clearly confirming an impaired cooperativity. Accordingly, T141A/G142A and G146A mutations, but not Q213A, significantly decrease the $Kd_1/K_{d2}$ value compared to wild-type RRM1–2 (*Figure 4c*). Through the NMR spectra analysis (*Figure 4d*, *Figure 4—figure supplement 2*), we also noticed that residues M202, E206, R208, M218, and V220, all located in the same region, display affected CSPs and/or peak broadenings when comparing T141A/G142A and G146A mutants with the wild-type RRM1–2.

Altogether, the biochemical and structural analysis point toward an interaction between the RRM1 loop 3 and a pocket around V220, in RRM2, which would enable the RRM1–2 dimerization on 12 GU repeats.

## The structural analysis of RRM1–2 self-interaction on long RNA revealed the interaction pocket

To obtain additional information about the intermolecular RRM1–2 interface in the presence of 12 GU repeats, we analyzed RRM1–2/RNA complexes in solution by small-angle X-ray scattering (SAXS) (*Figure 5a*). Size-exclusion chromatography analysis revealed the presence of monodisperse complexes with a size corresponding to a dimer (*Figure 5—figure supplement 1* and *Supplementary file 1*). The SAXS analysis further confirmed the formation of a complex indeed comprising two units of RRM1–2 proteins bound to 12 GU repeats. We then generated a molecular dynamics (MD) model of the RRM1–2 dimer in the presence of 12 GU repeats as described in Methods. The starting configuration consists of an RNA $(GU)_{12}$ oligonucleotide which binds to two RRM1–2 units in a 5′-to-3′ direction from RRM1 to RRM2 in the case of both two monomers. We also used an RNA linker of two nucleotides (*Lukavsky et al., 2013*), here GU, between RRM1 and RRM2 for each RRM1–2 monomer. The first monomer binds to this RNA from G in position 1 (G1) to G9, the U in position 10 remains free and the second monomer binds from G11 to G19 in the same configuration as the first monomer. The model shows how the protein dimer drives and stabilizes the RNA in a configuration displaying a bend at G13 with an average bending angle of 97.5° (*Figure 5—figure supplement 2*). The calculated plot deduced from the MD model fits the experimental SAXS curve with a $x^2$ value of 1.28 (*Figure 5a*). This model highlights an interaction interface created between a pocket around V220, located in the RRM2 domain of the first monomer and the RRM1 loop 3 of the second monomer (*Figure 5b*). The RRM2 pocket groove is built from residues belonging to the β2 strand (M218, D219, and V220) and the α1 helix (E204, D205, and R208). As observed in *Figure 5b* (left panel), the RRM2 pocket of the

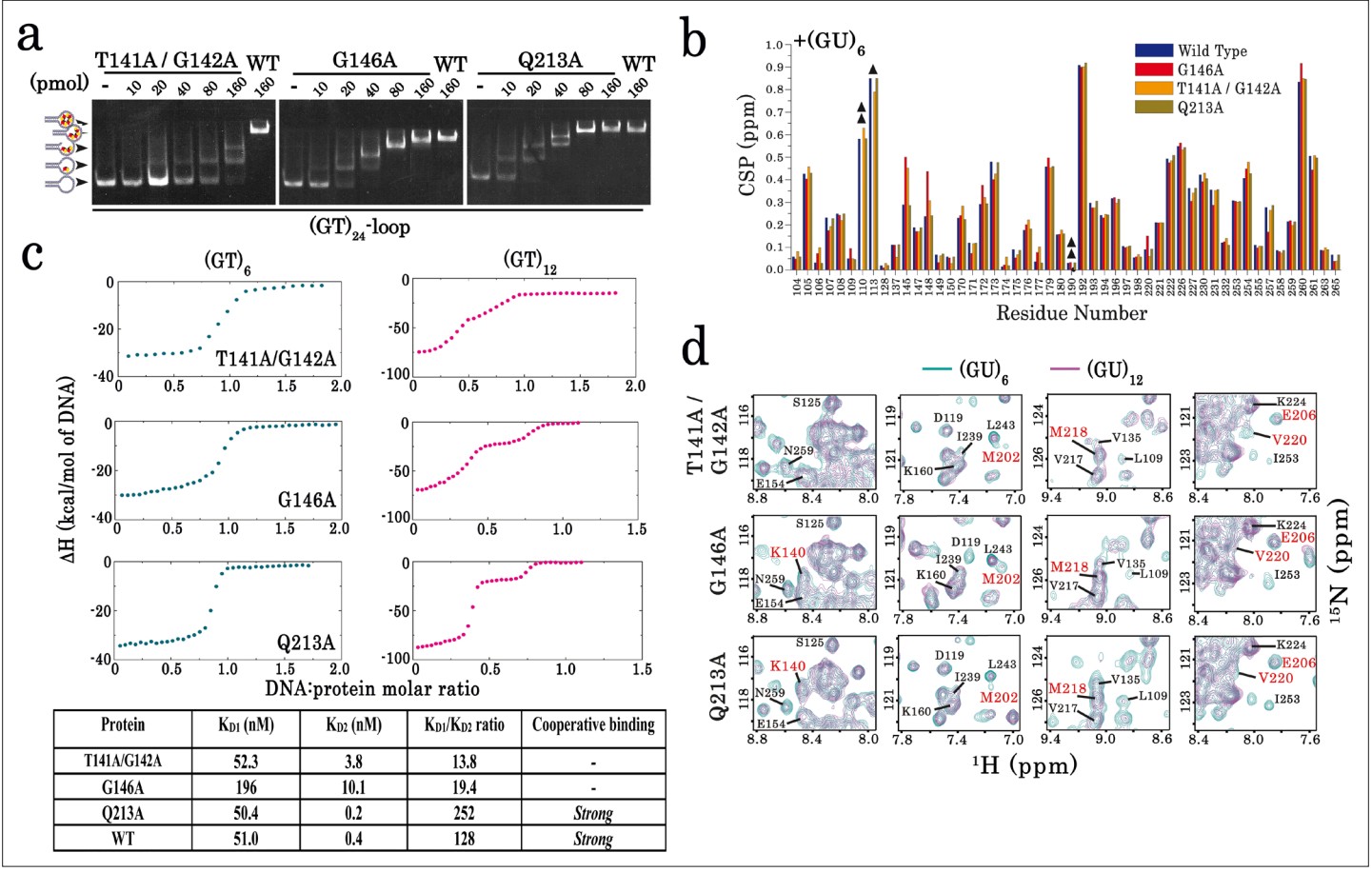

**Figure 4.** Characterization of TDP-43 mutants with an impaired cooperative binding to RNA. (**a**) EMSA experiments were performed by using increasing protein concentrations of T141A/G142A, G146A, and Q213A mutants and 10 pmol of a stem-loop DNA to assess multimerization changes on RRM1–2 (as in *Figure 1c*). Saturated amounts of wild-type RRM1–2 were used as a control (last lane). DNA-protein complexes are pointed out with head-arrows. (**b**) Plot showing CSPs for RNA-binding residues along RRM1–2 mutants bound to $(GU)_6$ compared to their free forms. In most cases, the binding to $(GU)_6$ RNA provokes CSPs comparable to the wild-type RRM1–2. ΔΔ, ambiguous assignment; Δ, signal vanishing. (**c**) The binding of RRM1–2 mutants to $(GT)_6$ or $(GT)_{12}$ oligonucleotides, was monitored by ITC. Whereas the ITC curves of wild type and Q213A are similar (see *Figure 1d*), the ITC curves related to T141A/G142A and G146A mutants decrease more continuously with less marked plateaus, reflecting an impaired cooperative binding to GT repeats. Lower panel: Kd1/Kd2 ratios obtained from ITC data. T141A/G142A and G146A have lower $Kd_1/Kd_2$ ratio values than wild type and Q213A, reflecting an impaired cooperativity. ITC statistics with thermodynamic parameters are indicated in *Supplementary file 4*. (**d**) Zoom in on the superimposed $^1H$-$^{15}N$ SOFAST-HMQC spectra (see full NMR spectra in *Figure 4—figure supplement 2*) of $^{15}N$-labeled RRM1–2 mutants bound to $(GU)_6$ (turquoise) or to $(GU)_{12}$ (magenta). The residues affected during wild-type RRM1–2 dimerization (see *Figure 2b,c*) are highlighted (red). Q213A mutant shows the same CSPs as the wild-type protein. However, in the case of T141A/G142A and G146A, we no longer detected the CSPs associated to dimerization.

The online version of this article includes the following source data and figure supplement(s) for figure 4:

**Source data 1.** EMSA experiments on T141A/G142A mutant (see legend of *Figure 4a*).

**Source data 2.** EMSA experiments on G146A mutant (see legend of *Figure 4a*).

**Source data 3.** EMSA experiments on Q213A mutant (see legend of *Figure 4a*).

**Source data 4.** NMR data for RNA-binding residues along RRM1–2 mutants bound to $(GU)_6$ compared to their free forms (see legend of *Figure 4b*).

**Source data 5.** ITC data obtained from the binding of RRM1–2 mutants to (GT)6 or (GT)12 oligonucleotides (see legend of *Figure 4c*).

**Figure supplement 1.** Raw calorigrams corresponding to the ITC curves displayed in *Figure 4c*.

**Figure supplement 1—source data 1.** ITC data obtained from the binding of RRM1–2 mutants to (GT)6 or (GT)12 oligonucleotides (see legend of *Figure 4—figure supplement 1*).

**Figure supplement 2.** NMR analysis of the interaction of TDP-43 mutant fragments with GU-repeats.

**Figure supplement 3.** Effect of point mutations on the stability of the monomeric and dimeric structures of wild type and mutants of RRM1–2 fragment.

**Figure supplement 3—source data 1.** Effect of point mutations on the stability of the monomeric and dimeric structures of wild type and mutants of RRM1–2 fragment (see legend of *Figure 4—figure supplement 3*).

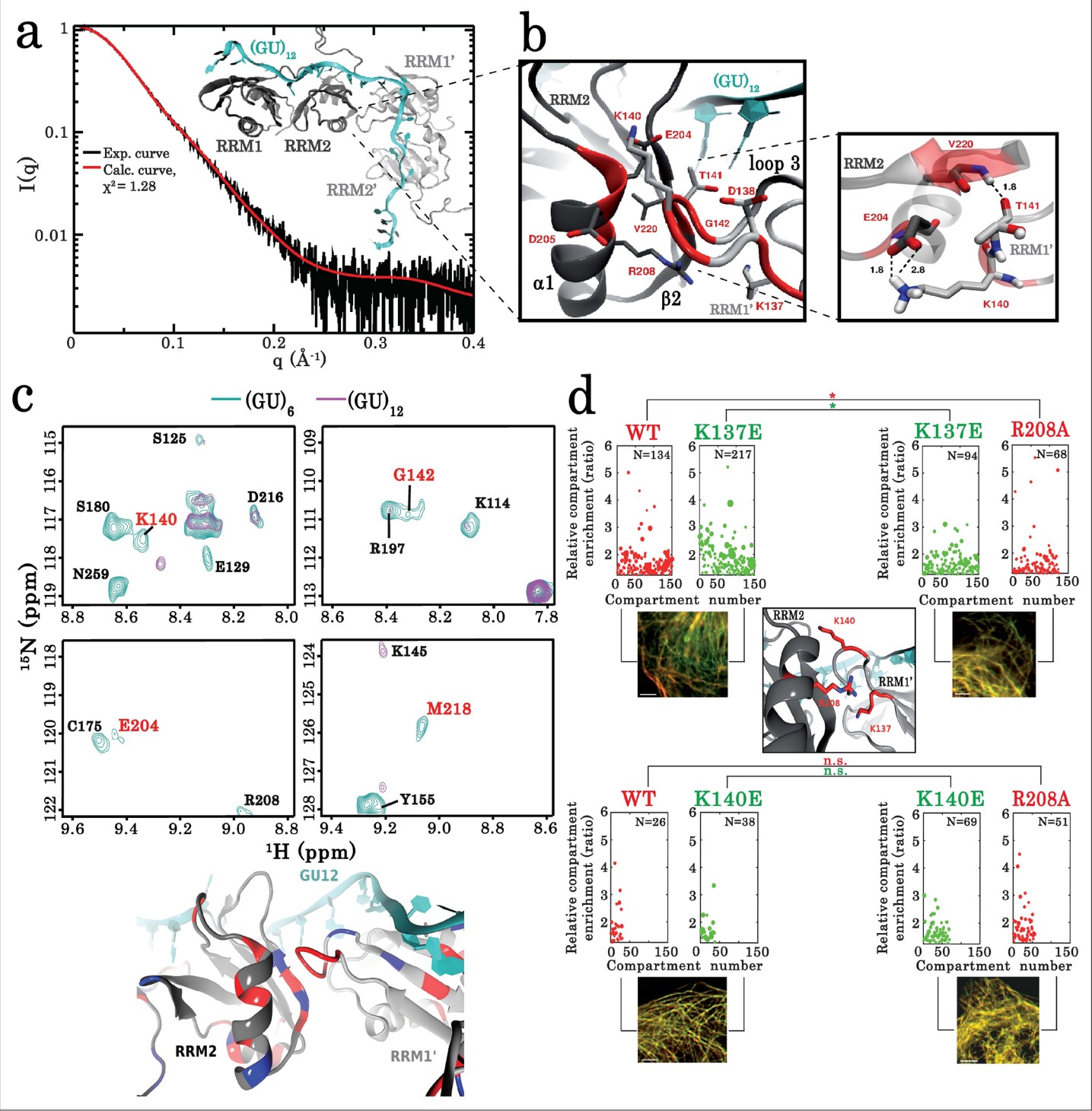

**Figure 5.** Key residues governing the RRM-dependent TDP-43 multimerization on RNA targets. (**a**) Superimposition of calculated (red curve) and experimental (black dots) SAXS curves corresponding to RRM1–2 bound to (GU)$_{12}$. SAXS curves were calculated from all-atoms model using the program GAJOE from the suite EOM. The corresponding $\chi^2$ values are indicated. The inset is a 3D representation of the model built using MD simulations from which the conformational state at equilibrium was considered. (**b**) Zoom in on the 3D model corresponding to the RRM1–2 bound to (GU)$_{12}$ showing the protein-protein interface created by the interaction of residues located in the $\alpha$-helix $\alpha 1$ and $\beta$-strand $\beta 2$ belonging to the RRM2 (first monomer) with residues located in the RRM1 loop 3 (second monomer). Interacting couples are highlighted in red and interaction bonds are shown by dotted lines. Numbers in black reflect the distance (Å). (**c**) *Upper panel* shows a zoom in on the superimposed $^1$H-$^{15}$N SEA-HSQC spectra (see full NMR spectra in *Figure 5—figure supplement 5*) of $^{15}$N-labeled RRM1–2 bound to (GU)$_6$ (turquoise) or to (GU)$_{12}$ (magenta). The residues present at RRM1–2 dimerization interface (highlighted in red) are no longer exposed to the solvent. *Lower panel* shows the global changes derived from SEA-experiment in

*Figure 5 continued on next page*

*Figure 5 continued*

solvent-exposed amides located at the protein-protein interface which are mapped on the 3D structure obtained from MD simulations (blue: exposed, red: not exposed). (**d**) As in *Figure 3c*, the microtubule bench assay was used to quantify the compartmentalization of different forms of TDP-43 co-expressed in HeLa cells. *Center panel:* view on the close proximity between R208 in RRM2 (first monomer) and K137 in RRM1 (second monomer). *Upper panel* shows a demixing phenotype between wild-type and K137E TDP-43. In contrast, R208 better mixes with K137E than wild-type TDP-43. *Bottom panel*, as a control, this behavior is not observed in the case of K140E. Scale bar: 10 µm. p<0.05*; p<0.01** (paired two-sample Kolmogorov–Smirnov test). n.s. non-significant. N, number of compartments.

The online version of this article includes the following source data and figure supplement(s) for figure 5:

**Source data 1.** Small-angle X-ray scattering (SAXS) data of RRM1–2 bound to (GU)$_{12}$ (see legend of *Figure 5a*).

**Source data 2.** The microtubule bench assay was used to quantify the compartmentalization of different forms of TDP-43 co-expressed in HeLa cells (See legend of *Figure 5d*).

**Figure supplement 1.** Small-angle X-ray scattering (SAXS) analysis of TDP-43 RRM1–2 fragment alone or bound to GU-repeats.

**Figure supplement 1—source data 1.** SAXS analysis of TDP-43 RRM1–2 fragment alone or bound to GU-repeats (See legend of *Figure 5—figure supplement 1a*).

**Figure supplement 1—source data 2.** SAXS analysis of TDP-43 RRM1–2 fragment alone or bound to GU-repeats (See legend of *Figure 5—figure supplement 1b*).

**Figure supplement 2.** Analysis of the kink angle stability.

**Figure supplement 2—source data 1.** Analysis of the kink angle stability (See legend of *Figure 5—figure supplement 2*).

**Figure supplement 3.** Interaction of RRM1–2 protein fragment with two different GU-rich oligonucleotides.

**Figure supplement 3—source data 1.** SAXS analysis of TDP-43 RRM1–2 fragment bound to AUG12-(GU)$_6$ oligonucleotide (See legend of *Figure 5—figure supplement 3*).

**Figure supplement 4.** Free energy landscape (FEL) of RRM1–2 dimer in complex with (GU)$_{12}$ sampled from 110 ns of MD simulation.

**Figure supplement 4—source data 1.** Ø Free energy landscape (FEL) of RRM1–2 dimer in complex with (GU)12 (See legend of *Figure 5—figure supplement 4*).

**Figure supplement 5.** Interaction of RRM1–2 protein fragment with GU-repeats.

first monomer accommodates the RRM1 loop 3 of the second monomer through several interaction pairs displaying the lowest free energy (K137-M218, K140-E204, K140-D205, T141-V220, G142-D219, and H143-D219) (*Supplementary file 2*). Among the amino acid residues constituting the RRM1 loop 3, K140 and T141 contribute to ca. 65 % of the total free energy provided by this loop to stabilize the interface (*Supplementary file 2*). In the RRM2 pocket, E204, D219, and V220 contribute to ca. 70% of the total free energy invested in the interface by this pocket (*Supplementary file 2*), which is stabilized by an intramolecular network of interactions involving two residues in the β2 strand (V217 and V220) and three residues of the α$_1$ helix (E204, L207, and R208) (data not shown). Noteworthy, by applying NMR and SAXS approaches together with MD calculations, the identified intermolecular interface is also observed in RNA/RRM1–2 complexes when using a different RNA oligonucleotide, 5′-GUGUGAAUGAAUGUGUGUGUGUGU-3′ (*Figure 5—figure supplement 3*).

The striking reappearance of V220 in the NMR spectrum of T141A/G142A mutant interacting with 12 GU repeats also points toward an interaction between the RRM2 pocket and RRM1 loop 3 (*Figure 4d*, *Supplementary file 2*). In turn, as suggested from G146A mutant investigations, the substitution of the G146 residue, most probably reduces the overall flexibility of the loop 3, thus impairing the multimerization process (*Figure 4c,d*, *Figure 4—figure supplement 2*, and *Figure 4—figure supplement 3*), while G146 has most probably no significant interaction with RRM2 residues of the first monomer by itself. To further delineate the intermolecular interaction in the RRM1–2 dimer, we assessed the accessibility of RRM1–2 residues to water molecules in presence of 6 or 12 GU repeats, the latter only leading to dimerization. The accessibility to water of unstructured residues K137, L139, K140, H143, and G142, located in the RRM1 loop 3, was significantly affected in the dimeric state but not in the monomeric form. Residues located in the RRM2 pocket around V220, E204, M218, and V220 itself, also display reduced accessibilities to the solvent when the dimer is formed supporting the MD model (*Figure 5c*, *Figure 5—figure supplement 5*).

The results of SAXS, NMR, and MD experiments thus point toward an interface between the RRM2 pocket around V220 and the RRM1 loop 3, when two RRM1–2 monomers interact with 12 GU repeats.

## Compensatory mutations further confirmed the TDP-43 intermolecular interface by using the MT bench assay

Given the lines of evidence pointing toward an interaction between the pocket around V220 in RRM2 of the first monomer and the RRM1 loop 3 of the second monomer, we devised a direct strategy to probe the relevance of this interaction in a cellular context. We noticed the close proximity of residues K137 and R208 in the dimer from the MD structure (*Figure 5d*, center panel). We then introduced an artificial electrostatic interaction between K137 and R208 through a single substitution of K137 by a glutamic acid residue. In the K137E mutant, a perturbing interaction between the introduced E137 residue with R208 in the wild-type TDP-43 may interfere with the functional RRM1–2 intermolecular interactions. Control experiments showed no significant demixing between wild-type and mutant K137A TDP-43 proteins on microtubules in cells (*Figure 3—figure supplement 1*). However, a strong demixing of wild-type and mutant K137E proteins was observed (*Figure 5d*, upper panel). The R208 residue was mutated into alanine, an uncharged residue, in order to prevent its putative intermolecular interaction with E137 (in K137E mutant). Interestingly, the K137E mutant better mixed with the R208A mutant than with wild-type TDP-43 (*Figure 5d*, upper panel). Furthermore, when K140 is replaced by a glutamic acid residue (K140E mutant), no effect was observed on the mixing with either wild-type TDP-43 or R208A mutant, most probably because E140 is too far away from R208 in the dimeric form (*Figure 5d*, bottom panel).

Therefore, these results support the spatial proximity between K137 and R208 in TDP-43 multimer, which is consistent with an intermolecular interaction between the RRM1 loop three and the V220 pocket in the RRM1–2 dimer.

## GT repeats increase the solubility of full-length TDP-43 in vitro but to a lesser extent the G146A solubility

After the identification of a cooperative binding of TDP-43 to GU and GT-repeats, we asked whether the TDP-43 multimerization on GU- or GT-repeats may antagonize the formation of full-length TDP-43 aggregates in vitro. An intermolecular interaction of the RRM2 pocket around V220 with RRM1 loop 3 of the second monomer could notably limit a head-to-tail assembly (*Wang et al., 2018b*) that is detrimental to TDP-43 solubility. The solubilities of full length TDP-43 and G146A mutant, which is defective in its cooperative association to GU- or GT-repeats, were then probed by sedimentation assays. We first noticed that TDP-43 at 10 µM was almost entirely found in the pellet after centrifugation, in agreement with its aggregation in the test tube (*Figure 6a*). These aggregates are detergent-insoluble (2 % of Triton X-100). The analysis of the higher order assembly of TDP-43 by Atomic Force Microscopy (AFM) also confirms the presence of granular TDP-43 structures in the absence of GT-repeats (*Figure 6c*). However, in the presence of increasing concentrations of 24 GT-repeats, a larger amount of TDP-43 was detected in the supernatant (three independent replicates). In addition, smaller TDP-43-rich structures than aggregates of TDP-43 alone, in the absence of GT-repeats, were detected by AFM. In agreement with the critical role of GT-repeats in preserving TDP-43 solubility, DNase treatment of pre-incubated TDP-43/GT-repeats complexes resulted in the appearance of spherical TDP-43 aggregates by AFM and a larger amount of TDP-43 in the pellet. Moreover, 6 GT-repeats that can bind only one TDP-43 monomer and 48 nt-long poly(dA) oligonucleotides for which TDP-43 has a lower affinity failed to solubilize full-length TDP-43 to the same extent as did 24 GT repeats under the same conditions (*Figure 6b*, *Figure 6—figure supplement 1*). When the solubility of G146A mutant was probed by sedimentation assays, 24 GT repeats increased the solubility of G146A but to a lesser extent than wild-type TDP-43. In agreement with this result, 24 GT repeats were less potent to limit the formation of granular structures of G146A compared to wild-type TDP-43 (see arrows, *Figure 6c*).

In contrast with wild-type TDP-43, the solubility of G146A TDP-43 mutant thus cannot be increased significantly in the presence of mRNA in vitro, most probably because of an impaired cooperative binding to mRNA.

## Cooperativity-defective mutants tend to form cytoplasmic mRNA-negative condensates

As TDP-43 cooperativity antagonizes the formation of TDP-43 aggregates in vitro, we explored whether the cooperative binding of TDP-43 to mRNA in cells plays a central role in the formation of reversible compartments. HA-tagged wild-type and mutant TDP-43 were therefore expressed in HeLa cells

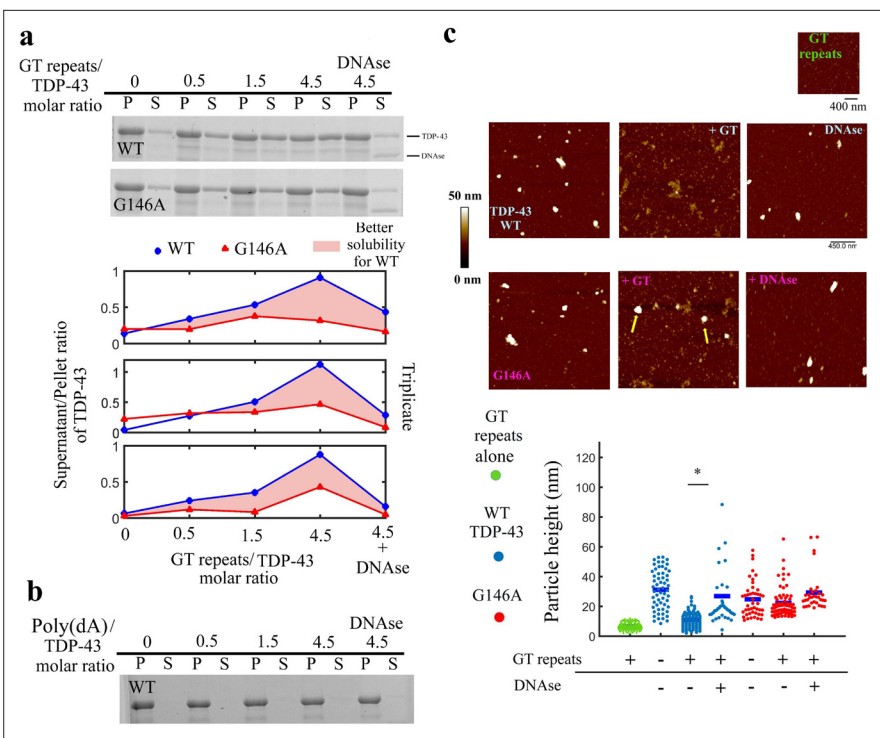

**Figure 6.** GT-repeats increase the solubility of full-length wild-type TDP-43 but to a lesser extent the solubility of G146A mutant. (**a**) *Higher panel:* Sedimentation assays of full length recombinant TDP-43 and the G146A mutant in the absence or presence of 24 GT repeats. Recombinant proteins were diluted in 20 mM Tris–HCl, pH 7.4 containing 25 mM KCl, 0.5 mM DTT, and 2 mM $MgCl_2$ (Buffer B) to a final concentration of 10 µM and incubated in the presence or absence of ssDNA for 5 min. The supernatant and pellet content after centrifugation were analyzed on SDS-PAGE and gels were stained with Coomassie blue. When indicated, the DNAse treatment was performed for 5 min after a 5 min preincubation of TDP-43 with GT-repeats. *Lower panel:* Analysis of the ratio of wild-type and mutant TDP-43 found in the supernatant and in the pellet. Quantification was performed using an Amersham Typhoon Imagers. Three independent experiments were performed (gels are shown in *Figure 6— figure supplement 1*). (**b**) Same as (**a**) with 48 nt-long Poly(dA) DNA and wild-type TDP-43. Poly(dA) DNA failed to increase TDP-43 solubility. (**c**) *Higher panel:* AFM images of the higher order assembly of full length wild-type or mutant TDP-43. Before their deposition onto a mica surface, indicated proteins (2 µM) were incubated for 5 min in the buffer B with or without 24 GT repeats (10 µM) (and DNAse when indicated). Arrows show the presence of spherical aggregates when G146A mutant was incubated with GT repeats. *Lower panel:* Quantification of the particle heights under the indicated conditions was performed with Bruker Nanoscope analysis software. p<0.05*; n.s. non-significant (paired t-test).

The online version of this article includes the following source data and figure supplement(s) for figure 6:

**Source data 1.** GT-repeats increase the solubility of full-length wild-type TDP-43 but to a lesser extent the solubility of G146A mutant (See legend of *Figure 6*).

**Figure supplement 1.** SDS-PAGE gels used for the quantification of fractions of full-length TDP-43 and G146A proteins found in the pellet and in the supernatant in sedimentation assays.

---

to analyze by optical microscopy observations (*Maharana et al., 2018*) their spatial distribution in cells. While wild- type TDP-43 and mutants displayed a homogenous distribution in the nucleus and cytoplasm in most cells, we noticed the presence of TDP-43 condensates in the cytoplasm of different morphology and mRNA composition in few cells. Wild-type and Q213A TDP-43 forms, which bind to GU-rich repeat cooperatively, were associated to large granules enriched in mRNA (*Figure 7a,b*, *Figure 7—figure supplement 1*). As these large condensates no longer appeared in the presence of cycloheximide, a stress granule inhibitor which loads ribosomes on mRNA, we concluded that TDP-43-rich condensates were reversible stress granules resulting from the expression of TDP-43. In contrast, G146A and K140A/ T141A form small and bright inclusions in the cytoplasm, but also in the nucleus. The G146A and K140A/ T141A inclusions are not sensitive to cycloheximide and poorly enriched in mRNAs.

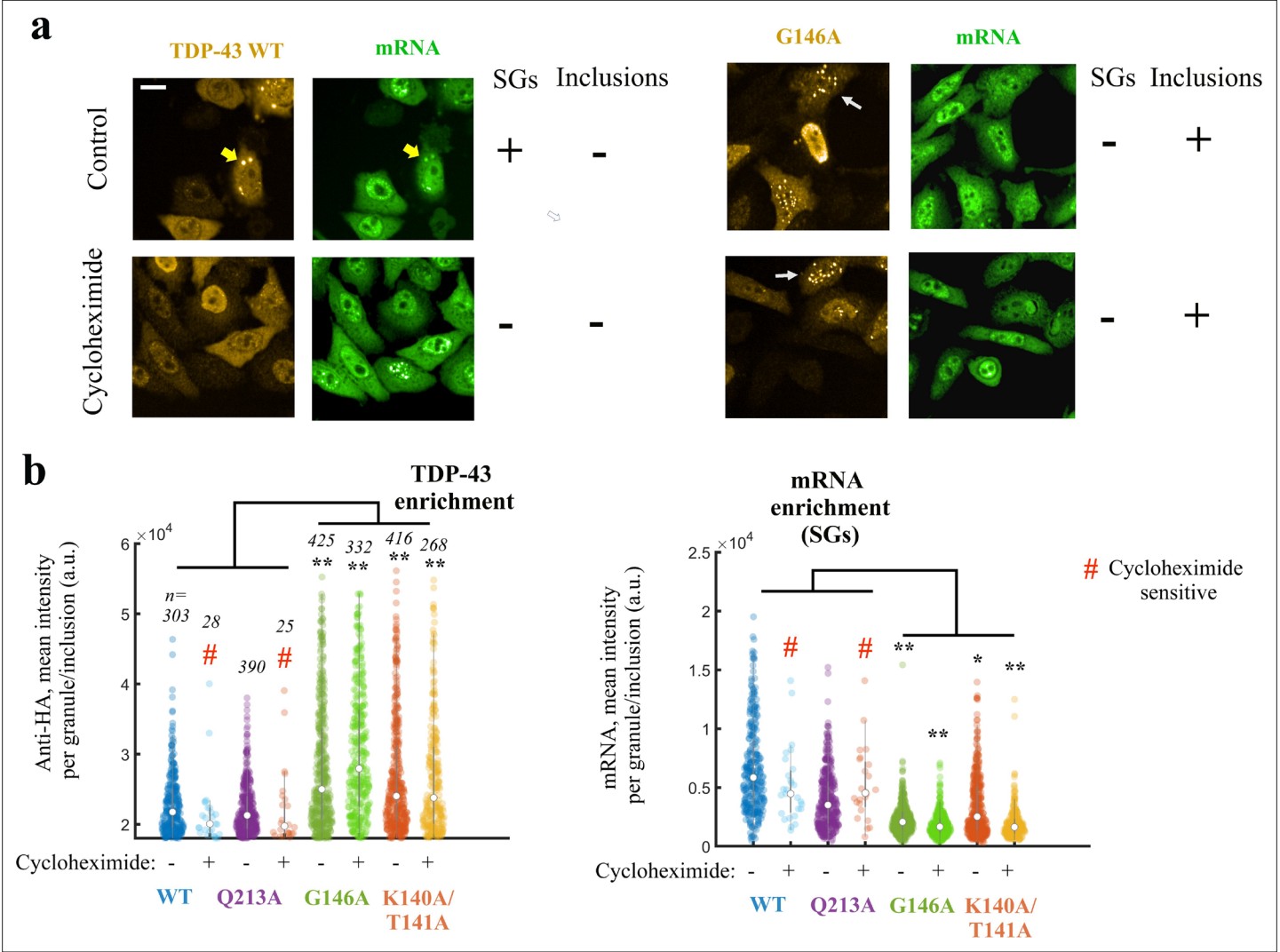

**Figure 7.** Expressing TDP-43 mutants with an altered cooperative binding mRNA in HeLa cells leads to the formation of mRNA-poor TDP-43 condensates in small fraction of cells. (**a**) Subcellular distribution of wild-type TDP-43 or G146A mutant. Cycloheximide treatment was used to dissociate stress granules in the cell cytoplasm, when indicated in the figure. Wild-type TDP-43 is generally homogenously distributed in the cytoplasm but can be also found in cytoplasmic mRNA-rich stress granules (yellow head-arrows). G146A mutant is also generally homogenously distributed in the cytoplasm but can be found in brilliant condensates in the cytoplasm and G146A condensates do not colocalize with mRNAs (white head-arrows). Scale bar: 40 μm. Representative images of larger areas are shown in *Figure 7—figure supplement 1b*. (**b**) Violinplots representing TDP-43 (anti-HA) and mRNA (in situ hybridization with poly(dT) probes) fluorescence intensity in the cytoplasmic granules/aggregates detected under indicated conditions. Wild-type TDP-43 or Q213A mutant, a negative control, can be recruited in mRNA-rich stress granules that disappeared after cycloheximide treatment. On the other hand, K140A/T141A and G146A mutants are located in dense cytoplasmic condensates poorly enriched in mRNAs. Interestingly, K140A/T141A and G146A are not sensitive to cycloheximide. Cytoplasmic granules/aggregates were detected automatically by using Cell Profiler. n: number of granules/aggregates detected. $p < 0.05^*$; $p < 0.01^{**}$ (paired two-sample t-test).

The online version of this article includes the following source data and figure supplement(s) for figure 7:

**Source data 1.** Expressing TDP-43 mutants with an altered cooperative binding to mRNA in HeLa cells leads to the formation of mRNA-poor TDP-43 condensates in small fraction of cells (See legend of *Figure 7b*).

**Figure supplement 1.** Cytoplasmic and nuclear G146A condensates are not stress granules.

Together, these results support a preserved solubility of wild-type TDP-43 and Q213A mutant and the formation of cytoplasmic inclusions for cooperativity-defective mutants, G146A and K140A/T141A.

To further probe the interplay by TDP-43 mutant with stress granules, we devised to use cellular stress conditions that induce a robust formation of stress granules. To this end, we used $H_2O_2$, an

oxidative stress agent, which, on its own, poorly induces stress granule assembly at the concentration used in this study (*Emara et al., 2012*) (300 µM, *Figure 8—figure supplement 1*) but used in combination with puromycin, that triggers premature chain termination during translation, generates a robust TDP-43-rich stress granule assembly in the cytoplasm (*Bounedjah al., 2014*, *Figure 8—figure supplement 1*). As a control, stress granules obtained in cells treated with both reagents form cytoplasmic stress granules of micrometric size (*Figure 8—figure supplement 1*) which are positive to stress granule markers G3BP-1, HuR, and FMRP-1. In mutations which do not affect TDP-43 cooperativity, G196A, Q213A, S258A/N259A as well as wild-type TDP-43, we detected the presence of TDP-43 in mRNA-rich stress granules and its homogenous distribution in the rest of the cytoplasm (*Figure 8—figure supplement 2*). However, the expression of cooperativity-deficient mutants, T141A/G142A and G146A, leads to the massive appearance of TDP-43-rich condensates in many cells, mostly in the nucleus (*Figure 8a,b*, and *Figure 8—figure supplement 2*). Importantly, cytoplasmic T141A/G142A and G146A condensates are distinct from stress granules and are poorly enriched in mRNAs (*Figure 8a–c*, *Figure 8—figure supplement 2*). These results clearly emphasize the role of cooperative binding of TDP-43 to mRNAs in the recruitment of TDP-43 in mRNA-stress granules. In agreement with this point, mutations in conserved residues that interact directly with RNA (*Lukavsky et al., 2013*), F149A, and R171A/D174A, also form mRNA-poor condensates in cells but not for W113A which affects the binding to mRNA to a lesser extent (*Figure 8—figure supplement 3*). The interaction of TDP-43 to cytoplasmic mRNAs whatever in or outside stress granule thus appears as critical to prevent the formation of mRNA-negative TDP-43 condensates.

## Cooperativity-defective mutants form insoluble condensates and display an altered nucleo-cytoplasmic shuttling after $H_2O_2$ treatment in cells

We then considered whether cellular condensates of cooperativity-deficient mutants were reversible or insoluble which would qualify them as protein inclusions. To probe whether the assemblies of TDP-43 mutant are dynamic or irreversible, cycloheximide and Triton X-100, a detergent, were used to dissolve TDP-43 condensates. While cycloheximide and Triton X-100 successfully dissociate stress granules, even those enriched with overexpressed wild-type TDP-43, both treatments failed to dissolve nuclear and cytoplasmic aggregates of G146A mutant (*Figure 8—figure supplement 3*). In agreement with this results, G146A cytoplasmic condensates do not recruit other stress granules proteins, G3BP-1, HuR, and FMRP-1, in contrast to wild-type TDP-43 (*Figure 8—figure supplement 3*). In addition, Sam68, a marker of nuclear stress granules (NSGs, *Denegri et al., 2001*), is not recruited in nuclear G146A condensates and formed distinct Sam68 nuclear bodies (SNBs), which means that nuclear G146A condensates cannot be considered as NSGs (*Figure 7—figure supplement 1*).

Given that $H_2O_2$ promotes the nucleocytoplasmic shuttling of TDP-43 in HeLa cells (*Figure 8—figure supplement 1*, *Singatulina et al., 2019*), we asked whether an altered TDP-43 translocation may provide a rational explanation for their sequestration in the nucleus and possibly their aggregation (*Figure 8b,d*). As shown in *Figure 8*, TDP-43 inclusions with G146A, K140A/T141A, and T141A/G142A mutations are indeed mostly nuclear after hydrogen peroxide treatment (*Figure 8a, b,d*). However, even when condensates of TDP-43 mutants are not present in cells, mutations interfering with the cooperative binding of TDP-43 to mRNA still significantly reduces the shuttling of TDP-43 from the nucleus to the cytoplasm (*Figure 8—figure supplement 1*). TDP-43 translocation after $H_2O_2$ treatment may thus rely on an intact cooperative binding of TDP-43 to mRNA.

## Discussion

For all researchers working with the recombinant TDP-43 in vitro, the solubility of full-length TDP-43 is an issue. Two interactions have been associated with homotypic TDP-43 interactions leading to TDP-43 self-assembly in the test tube. First, the weak and multivalent interactions between the long and unstructured LCD mediate liquid–liquid phase separation (*Conicella et al., 2016*; *Conicella et al., 2020*; *Shin and Brangwynne, 2017*). Second, the dimerization of the N-terminal domain is associated to a head-to-tail aggregation of TDP-43 (*Wang et al., 2018a*; *Afroz et al., 2017*). While TDP-43 self-attraction also occurs under physiological conditions in cells (*Afroz et al., 2017*), the reversibility and

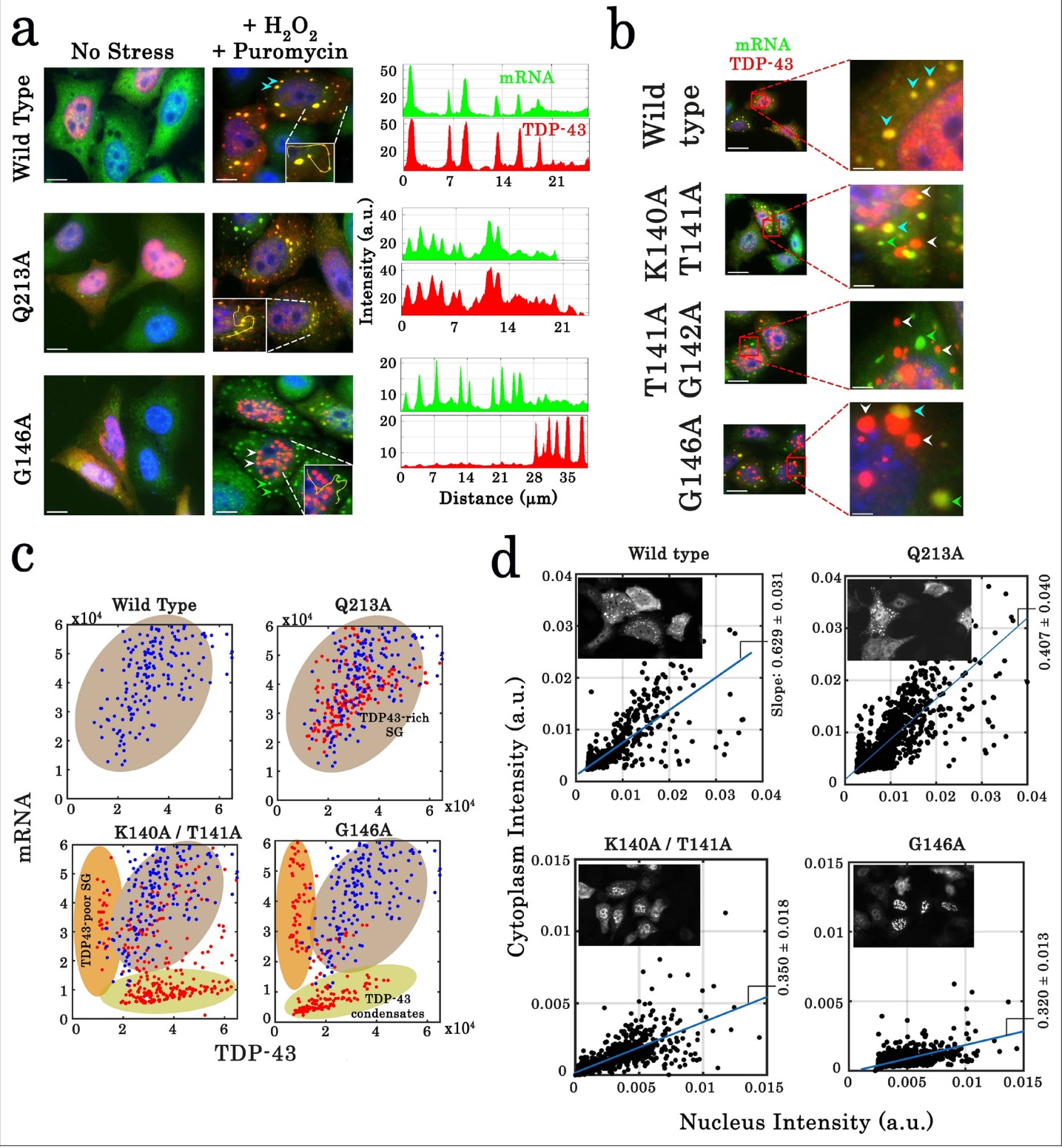

**Figure 8.** Cellular stress promotes the formation of condensates for cooperativity-defective mutants in cells that are distinct from stress granules. (**a**) Subcellular distribution of wild-type TDP-43 or mutants (Q213A and G146A) upon H₂O₂/puromycin treatment. Wild-type TDP-43 and Q213A are found in cytoplasmic mRNA-rich stress granules (cyan head-arrows). G146A generates the formation of mostly nuclear TDP-43 condensates coexisting with TDP43-poor stress granules in the cytoplasm (white head-arrows). Scale bar: 10 μm. *Right panels:* line profiles of TDP-43 (red) and mRNA (green) intensities along the indicated path. (**b**) Cytoplasmic distribution of wild-type TDP-43 or mutants (K140A/T141A, T141A/G142A, and G146A) under stress conditions. Wild-type TDP-43 is only present in mRNA-rich stress granules (cyan head-arrows) but mutants also formed mRNA-poor TDP-43 condensates

*Figure 8 continued on next page*

*Figure 8 continued*

(white head-arrows) or mRNA-rich/TDP43-poor condensates (green head-arrows). (**c**) Statistical analysis of the subcellular compartments made of mRNA and TDP-43 in HeLa cells expressing wild-type TDP-43 or its mutants (K140A/T141A, T141A/G142A, G146A, and Q213A). Data from wild type (blue dots) are superimposed to data from mutants (red dots) in order to delineate formed subcellular compartments, TDP43-rich stress granules (brown), TDP43-poor stress granules (orange) and TDP43-condensates (green). Compartments were detected automatically using Cell Profiler. More than 100 compartments were analyzed for each condition. (**d**) Images of transfected HeLa cells (as displayed in the inset) were used to quantify the cytoplasmic and nuclear distribution of TDP-43 after cellular stress. The average slope for each distribution was then computed (blue line). For wild-type and Q213A mutant, the slope shows that TDP-43 is partly nuclear and cytoplasmic after its stress-induced nucleocytoplasmic shuttling. In contrast, for K140A/T141A and G146A mutants, the reduced slope indicates a nuclear retention of TDP-43. Cell cytoplasm and nucleus were detected automatically by using Cell Profiler. $N_{cell} > 150$.

The online version of this article includes the following source data and figure supplement(s) for figure 8:

**Source data 1.** Statistical analysis of the subcellular compartments made of mRNA and TDP-43 in HeLa cells expressing wild-type or TDP-43 mutants (See legend of *Figure 8c*).

**Source data 2.** Assessment of the cytoplasmic and nuclear distribution of TDP-43 after cellular stress (See legend of *Figure 8d*).

**Figure supplement 1.** The cooperative binding of TDP-43 to mRNA is critical for TDP-43 translocation.

**Figure supplement 1—source data 1.** Assessment of TDP-43 translocation (See legend of *Figure 8—figure supplement 1*).

**Figure supplement 2.** Effect of mutations on subcellular distribution of TDP-43 in HeLa cells.

**Figure supplement 2—source data 1.** Effect of mutations on subcellular distribution of TDP-43 mutants in HeLa cells (See legend of *Figure 8—figure supplement 2*).

**Figure supplement 3.** G146A mutation leads to the formation of insoluble condensates.

**Figure supplement 3—source data 1.** G146A mutation leads to the formation of insoluble condensates (See legend of *Figure 8—figure supplement 3c*).

dynamics of the formed multimers is generally preserved. The idea of a high RNA concentration in the nucleus which would prevent the formation of large assemblies of RBPs such as FUS and TDP-43 has been already well established (*Maharana et al., 2018*). RNA may indeed buffer high-order assemblies of RBPs at high concentration in vitro by dispersing proteins thus reducing the occurrence of self-interactions. However, whether this mechanism takes place in the nucleus remains unclear. The general high-level protein occupancy along mRNA sequences in cells (*Baltz et al., 2012*) may not allow free mRNA to buffer TDP-43 in the nucleus.

To decipher the mechanisms by which mRNA keeps TDP-43 soluble, we noticed that the binding of RBPs to mRNA is polarized (*Lukavsky et al., 2013*). In the case of TDP-43, RRM1 and RRM2 bind the 5′ side and the 3′ side, respectively. If we consider that multimers of TDP-43 are resulting from the concomitant and adjacent bindings of TDP-43 along mRNA, the orientation of TDP-43 therefore ensures that consecutive N-terminal domains can hardly dimerize and that consecutive LCD domains are spatially separated to reduce multivalent interactions. Thus, the hypothesis by which the polarized and cooperative binding of TDP-43 to mRNA would contribute to prevent TDP-43 aggregation does therefore make sense. In addition, cooperativity can be essential as the binding of TDP-43 to mRNA facilitates the binding of the next TDP-43 unit nearby whereof preventing a long and flexible RNA linker between two consecutive TDP-43 to accommodate a head-to-tail TDP-43 multimer structure. Here, we present a compelling evidence of a cooperative binding of TDP-43 RRM1–2 to long GU repeats. Based on an integrative structural analysis combining NMR spectroscopy, SAXS and MD, we have also identified the intermolecular RRM1–2 interface responsible for the cooperativity process. This intermolecular interface, which occurs upon RNA binding, is formed through an interaction network involving on one hand the β2 strand and α1 helix from RRM2 of the first monomer constituting a pocket, and on the other hand, the RRM1 loop 3 of the second monomer. Through our mutation screening, we notably identified a single mutation, G146A, which prevents the cooperative binding of RRM1–2 to GU repeats in vitro while preserving the affinity of RRM1–2 to RNA. In vitro, we also found that 24 GT repeats can preserve TDP-43 solubility in agreement with the buffering activity of RNA (*Maharana et al., 2018*). Interestingly, the increased TDP-43 solubility observed in the presence of 24 GT repeats was significantly reduced for the G146A mutant, which suggests a link between TDP-43 solubility and its cooperative binding to RNA. In cells, mutations impeding the cooperative binding of RRM1–2 to mRNA strikingly leads to the appearance of TDP-43 condensates mostly located in the nucleus under oxidative stress conditions (*Figure 8a*). The condensates of TDP-43 mutants, whatever

nuclear or cytoplasmic, are poorly enriched in mRNA, reflecting that mRNA could no longer prevent TDP-43 self-attraction. We hypothesized that the absence of mRNA is not due to the lower affinity of TDP-43 mutants for short RNA, as demonstrated by ITC and NMR experiments, but to an impaired cooperative binding of TDP-43 to long RNAs (*Figure 4*).

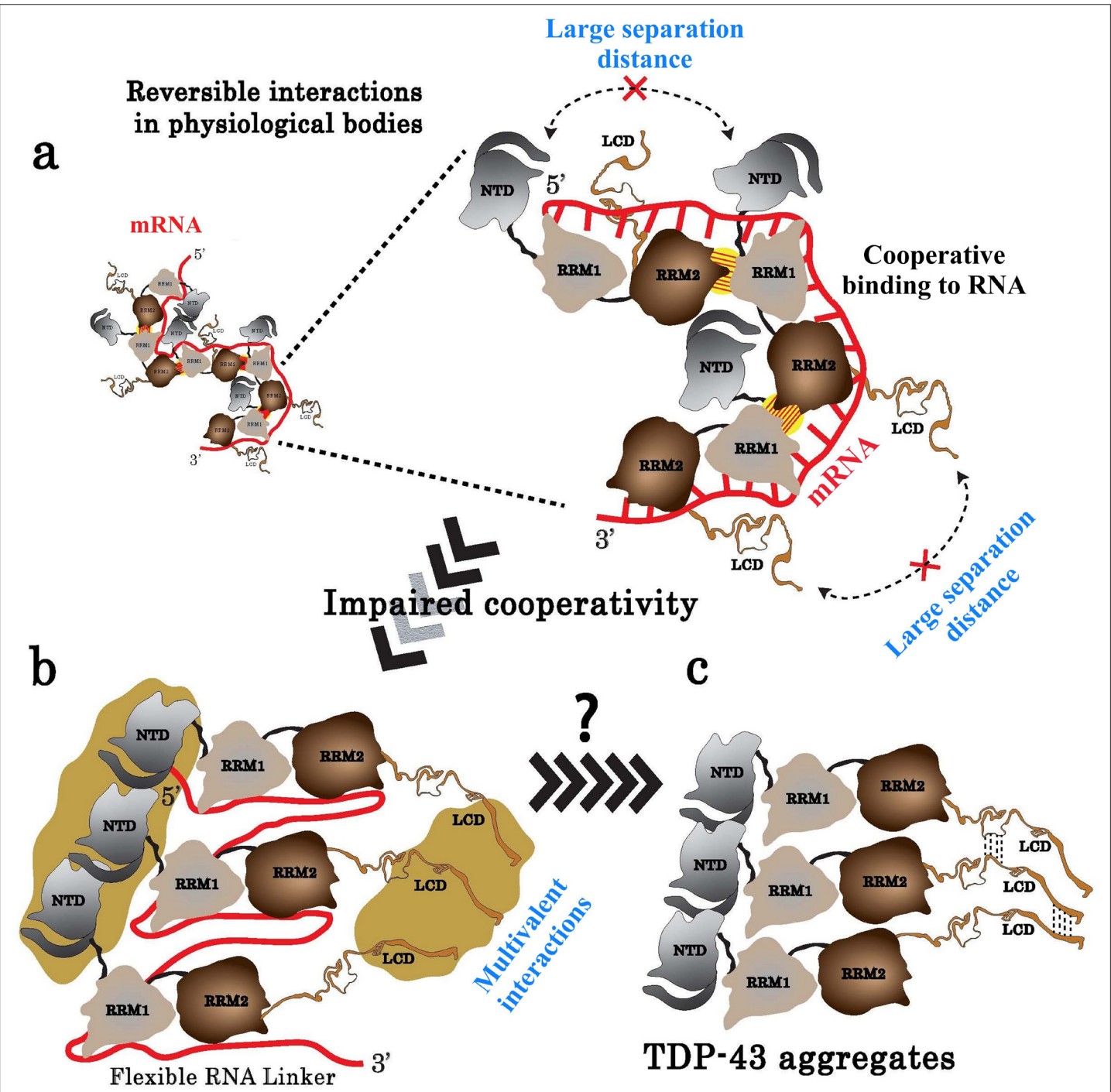

**Figure 9.** Model of TDP-43 assemblies in the presence of mRNA. (**a**) Under physiological conditions, an inter-protein interaction (yellow with hatches) involving RRM2 and RRM1 prevents self-attractions of both the N-terminal domain (NTD) and the low complexity domain (LCD) between adjacent TDP-43. (**b**) When TDP-43 assemblies are no longer stabilized by the RRM-dependent inter-protein interaction, the presence of a RNA linker separating two consecutive TDP-43 leaves room for multivalent interactions between NTDs and LCDs, rendering possible a head-to-tail assembly of TDP-43. (**c**) The reduced access of mRNA to RRMs in the head-to-tail assembly may explain the transition towards a mRNA-free TDP-43 aggregation.

The results presented here are in agreement with a mechanistic model based on an ordered assembly of TDP-43 mediated by its RRMs along its mRNA targets, which could constitute a steric barrier limiting short range self-interactions between consecutive N-terminal domains and LCDs. TDP-43/mRNA complexes therefore preserve their dynamics and solubility (*Figure 9*). However, this model does not exclude self-interactions between non-consecutive TDP-43 involving N-terminal domains (*Jiang et al., 2017*) or LCDs, in physiological conditions, to generate the formation of mRNA-rich dynamic assemblies in the nucleus (*Figure 9*). An impaired cooperativity may promote an anarchic attachment of TDP-43 along mRNA leading to an increased occurrence of interactions between the N-terminal domains and LCDs, which may in turn promote the head-to-tail aggregation of TDP-43. As the compact head-to-tail assembly leaves little room for the dynamic interaction of RRM1–2 with mRNA, a transition toward the formation of mRNA-poor TDP-43 condensates could be facilitated (*Figures 9 and 8b*). However, as recently shown, the chaperone activity of HSP70 could prevent RNA-free TDP-43 aggregation to form reversible anisosomes (*Yu et al., 2021*). The relative contribution of HSP70 and mRNA chaperone activities to prevent TDP-43 aggregation should thus deserve further investigations. In addition, TDP-43 phosphorylation events, notably those occurring in the C-terminal domain, were detected in the cytoplasmic TDP-43 inclusions in brains of ALS patients (*Hasegawa et al., 2008*). Phosphorylation in RRM domains may also occur to regulate the binding to GU-rich RNA and the splicing of introns (*Li et al., 2017*). It would be of interest to probe whether specific TDP-43 phosphorylation events interfere with its cooperative association to mRNAs and then up- or down-regulate TDP-43 aggregation in cells.

Regarding pathological mutations found in TDP-43 (*Van Deerlin et al., 2008*), most of them lie in the C-terminal low complexity domain and the three located in RRM-1 and -2 (P112H, D169G, and K181E) are not directly involved in the cooperative binding of TDP-43. The dramatic phenotypes leading to neurodegeneration only appear after aging. We may speculate that interfering strongly with cooperativity may thus alter the early stages of neuron development. However, we cannot exclude that some pathological mutations in RRM-1 and -2 may slightly interfere with the cooperative binding of TDP-43 to promote neurodegeneration. Additional work is needed to document this point. Importantly, whatever the link between pathological mutations and TDP-43 cooperativity, strategies aiming to interfere with the cooperative binding of TDP-43 to mRNA could be considered to reduce a risk of developing neurodegenerative diseases in adult patients.

Besides its role in preventing TDP-43 aggregation in cells, the cooperative binding of TDP-43 to mRNA may have physiological functions, consistent with the clusters of TDP-43 detected by CLIP analysis along intronic sequences. The cooperativity may enable to direct TDP-43 on long mRNA sequences, particularly for the processing of long introns (*Tollervey et al., 2011*; *Polymenidou et al., 2011*). Analyzing the cooperative behavior of TDP-43 on other sequences than pure GU-repeats could be of interest to understand the formation of intronic TDP-43 clusters in cells (*Bhardwaj et al., 2013*). In addition, understanding TDP-43 cooperativity may provide a structural basis for an RNA chaperone activity (*Ishiguro et al., 2017*), possibly to prevent the formation of UGGAA RNA foci in microsatellite expansion disorders (*Ishiguro et al., 2017*) and, more generally, to reduce the occurrence of pathogenic RNA–RNA interactions in introns. In addition to homotypic interactions among TDP-43 proteins, TDP-43 could also interact with other RBPs along introns which themselves have domains of low complexity. In a more general framework, it is thus possible that homotypic and heterotypic cooperativities between RBPs may preserve RBP solubility, which may be required to ensure the proper processing of mRNA and to avoid pathological aggregations.

## Materials and methods
### Protein production and purification

The recombinant His$_6$-tagged RRM1, RRM2, and RRM1–2 fragments from human TDP-43 were over-expressed in *E. coli* strain BL21 (DE3).

Cells carrying plasmids, pTDP-RRM1_101–192, pTDP-RRM2_176–277, and pTDP-RRM1-2_101–277 (encoding for the recombinant His$_6$-tagged RRM1, RRM2, and RRM1–2 fragments from human TDP-43, respectively) were grown at 37 °C in 2YT-ampicillin medium (1 L culture) (non-labeled proteins) or in minimal medium M9 supplemented with $^{15}$NH$_4$Cl (labeled proteins). When the optical density of the culture reached 0.7, IPTG was added at a final concentration of 1 mM, and growth was continued for

3 hr. Cells were harvested and washed with 20 mL of cold 25 mM Tris–HCl buffer, pH 7.4, containing 1 mM TCEP, 1 mM PMSF, and EDTA-free protease inhibitor Cocktail (Roche) and 1.5 M KCl (buffer A). The cell pellet (4.5 g wet wt) was suspended in 10 mL of the same buffer, and cells were disrupted by sonication on ice (Bioblock Vibracell sonicator, model 72412). The resulting suspension was centrifuged at 4 °C for 30 min at 150,000× g in a TL100 Beckman centrifuge. The supernatant was used for purification experiments.

TDP-43 fragments were purified basically following the manufacturer's recommendations (Qiagen). Imidazol (10 mM) was added to soluble fractions described above and incubated for 2 hr at 4 °C with Ni$^{2+}$-NTA-agarose (Qiagen) (20 mg of proteins/mL of resin) pre-equilibrated in buffer A. After incubation, the resin was transferred to an Econo-Pac chromatography column (Bio-Rad). The polymer was then washed extensively with buffer A containing 20 mM imidazole. The elution of the protein was obtained by increasing step by step the concentration of imidazole, from 40 to 250 mM, in buffer A. Pure protein-containing fractions (100–250 mM imidazole) were pooled and incubated with a His$_6$-tagged TEV protease to cleave off the His$_6$-tag peptide from the target protein. The protease (15 μg) was mixed with 1 mg of target protein (0.5 μM TEV to ~30 μM protein) in buffer A containing of 1 mM DTT and 1 mM EDTA. All digestions were conducted for 16 hr at room temperature. A PD-10 column (GE Healthcare) was used to remove imidazole and to exchange buffer. Then, TEV protease and His$_6$-tag peptide from target protein were trapped on Ni-NTA agarose column and target protein was recovered in pass-through (nonbinding) fraction. The protein was concentrated to 2 mL and conserved in 20 mM Tris–HCl buffer, pH 7.4, containing 25 mM KCl and 1 mM TCEP by using a PD-10 column. For NMR experiments, the $^{15}$N-labeled proteins were stored in phosphate buffer 15 mM pH 6.8 containing 25 mM KCl and 1 mM TCEP by using a PD-10 column (GE Healthcare). The final preparations were stored at –80 °C.

Full-length forms of TDP-43, wild-type, and G146A mutant were purified as previously described for full-length wild-type form (*Singatulina et al., 2019*). BL21(DE3) *E. coli* cells carrying plasmid pET-WT_TDP-43 and pET-G146A_TDP-43, were grown at 37 °C in 2xYT-ampicillin medium (1 L culture). When the optical density of the culture reached 0.7 OD at 600 nm, IPTG was added to a final concentration of 1 mM, and growth was continued for 3 hr. Cells were harvested and washed with 20 mL of cold buffer A. The cell pellet was suspended in 10 mL of the same buffer, and cells were disrupted by sonication on ice (Bioblock Vibracell sonicator, model 72412). The resulting suspension was centrifuged at 4 °C for 30 min at 150,000× g in a TL100 Beckman centrifuge. Then, the pellet was resuspended in 10 mL of the buffer A containing 6 M Urea, and 10 mM Imidazole and incubated for 120 min at 4 °C. The resulting suspension was centrifuged at 4 °C for 30 min at 200,000× g in a TL100 Beckman centrifuge. The supernatant was stored at −20 °C and used for purification experiments. The His6-tagged proteins were purified as follows: soluble fractions described above were incubated for 2 hr at 4 °C with Ni2+-NTA-agarose (15 mg of protein/mL of resin) pre-equilibrated in buffer A containing 6 M Urea, and 10 mM Imidazole. After incubation, the resin was transferred to an Econo-Pac chromatography column (Bio-Rad). The resin was then washed extensively with buffer A containing 6 M Urea, and 20 mM imidazole and elution of the protein was obtained by increasing step by step the concentration of imidazole, from 40 to 250 mM, in buffer A. The purity of the resulting protein preparations was monitored at all stages by SDS–PAGE. The pure protein-containing fractions (100–250 mM imidazole) were concentrated to 2 mL and then dialyzed overnight against 100 volumes of buffer containing 20 mM Tris–HCl, pH 7.4, 0.5 mM DTT, 6 M urea, and 200 mM NaCl. The final preparations were stored at −20 °C.

Site-directed mutagenesis of the *RRM1–2* coding gene from human *TDP-43* was carried out directly on the pTDP-RRM1-2_101–277 expression plasmid by using the 'Quikchange II XL site-directed mutagenesis kit' from Stratagene and appropriate oligonucleotides (Eurofins Genomics). The introduced mutations were checked by DNA sequencing (Eurofins Genomics). Overexpression and purification of mutated forms were performed following the method described above.

The protein purity was monitored at all stages of the purification by SDS–PAGE (*Figure 1—figure supplement 1*).

## Gel mobility shift assay

Increasing amounts of previously purified RRM1, RRM2, or RRM1–2 protein fragments were incubated in the presence of 10 pmol of the oligonucleotide [GATATAGAGGTAAGATAG-(GT)$_{24}$-CTATCTTACCTCTATATC]

expected to form a stem-loop structure (named $(GT)_{24}$-loop) with 24 GT dinucleotide repeats $(GT)_{24}$ being the loop. The stem structure facilitates the ethidium bromide visualization on a poly-acrylamide gel. Briefly, the mixtures were incubated in 10 µL of binding buffer (20 mM HEPES, pH 7.6 containing 25 mM KCl, 1 mM TCEP, and 2 mM $MgCl_2$) at room temperature for 20 min. Free and bound-to-protein oligonucleotides were separated in a 10 % poly-acrylamide gel in 0.5× TAE buffer at 80 V for 75 min on ice. Finally, gels were stained with 0.5 µg/mL of ethidium bromide.

### Cross-linking assay

Purified RRM1, RRM2, or RRM1–2 protein fragments were incubated in the presence of 10 pmol of $(GT)_{24}$-loop oligonucleotide, in the binding buffer. The mixtures were incubated at 37 °C during 30 min. Then, samples were subjected to cross-linking reaction using the BS3 chemical arm (bis(-sulfosuccinimidyl)suberate) (2 mM) at room temperature for 1 hr. The reaction was stopped by the addition of 100 mM Tris–HCl, pH 8.0, in order to neutralize the excess of BS3, and incubated in gentle agitation during 30 min at room temperature. When needed, cross-linked proteins were treated with benzonase (ThermoFisher Scientific) 0.2 µg/µL during 45 min at 32 °C. Finally, reaction mixtures were separated on SDS–PAGE (12 % poly-acrylamide) followed by visualization using Coomassie-blue.

### ITC measurements of protein/oligonucleotide binding

ITC experiments were carried out at 25 °C with a MicroCal PEAQ-ITC isothermal titration calorimeter (Malvern Instruments). All protein samples were dialyzed against the same buffer (15 mM phosphate buffer, pH 6.8, containing 25 mM KCl and 1 mM TCEP). The protein concentration in the microcalorimeter cell (0.2 mL) varied from 14 to 16 µM. In total, 36 injections of 1 µL (or 39 injections of 1 µL) of oligonucleotide solution (concentration from 80 to 150 µM) were carried out at 90 s intervals, with stirring at 650 rpm. Data were analyzed by using the Microcal PEAQ-ITC Analysis Software.

### Nuclear magnetic resonance on TDP-43 fragments

NMR samples: Purified $^{15}N$-labeled RRM1, RRM2, or RRM1–2 protein fragments were incubated with GU-rich RNA oligonucleotides (Eurogentec) containing 3, 6, or 12 GU repeats (named $(GU)_3$, $(GU)_6$, and $(GU)_{12}$, respectively) during 10 min at 25 °C. Free and RNA-bound protein samples were prepared in NMR buffer (15 mM phosphate, pH 6.8, containing 25 mM KCl and 1 mM TCEP) supplemented with SUPERase·In RNase Inhibitors (ThermoFisher Scientific). All samples were prepared in a final volume of 60 µL using 1.7 mm diameter capillary tubes (Bruker) and 2,2-dimethyl-2-silapentane-5-sulfonic acid as external reference in pure $D_2O$ (Eurisotop) for chemical shift referencing.

NMR measurements: NMR spectra were acquired on a Bruker AVIII HD 600MHz spectrometer equipped with a triple-resonance cryoprobe. For SOFAST-HMQC experiments (*Schanda and Brutscher, 2005*), resonances were obtained after 6 hr of acquisition at 298 K. The number of dummy scans and scans was respectively set to 16 and 512. Data were acquired with 2048 points along the direct dimension and with 128 $t_1$ increments with a relaxation delay of 0.2 s. Shaped pulse length and power were calculated by considering an amide $^1H$ bandwidth of 4.5 ppm and a chemical shift offset of 8.5 ppm. For SEA-HSQC experiments (*Lin et al., 2002*) employing a 5 KHz CLEANEX-spinlock, data were obtained at 310 K after 26 hr of acquisition implying 2048 and 128 points in the direct and indirect dimensions, 16 dummy scans and 256 scans with a relaxation delay of 2.5 s. Spectral widths were set to 12.5 ppm (centered at 4.7 ppm) in the $^1H$ direction and 34 ppm (centered at 115 ppm) in the $^{15}N$ dimension. Data were finally processed with Topspin 3.5 (Bruker).

NMR assignment: $^1H$ and $^{15}N$ chemical shifts of RRM1–2 residues were assigned using previous assignments obtained for the unbound RRM1 and RRM2 (BMRB Entries: 18,765 and 19,922, respectively) (*Chang et al., 2013*) and transferred to assign those of RRM1–2 in the unbound form. To assign spectra corresponding to $(GU)_6$-bound RRM1–2, chemical shifts were deduced from the previously assigned (AUG12)-bound RRM1–2 (BMRB Entry: 19290) (*Lukavsky et al., 2013*).

### Molecular dynamics (MD) simulations of free and RNA-bound RRM1–2 from TDP-43

For MD simulations, 10 systems were constructed: (1) RRM1–2 monomer in its free form; (2) RRM1–2 bound to $(GU)_3$; (3) RRM1–2 (wild type, Q213A, G146A, or T141A/G142A) bound to $(GU)_6$; and (4) RRM1–2 (wild type, Q213A, G146A, or T141A/G142A) bound to $(GU)_{12}$. The starting coordinates are

taken from the NMR structure of RRM1–2 monomer in complex with UG-rich RNA (AUG12) (PDB ID 4BS2) (*Lukavsky et al., 2013*).

System preparation and MD setup: First, the protein sequence was adjusted to the primary sequence used in the present work from GSH-Q101- to G277 (length 180 a.a). Second, for the RRM1–2/RNA complexes, the RNA sequence (AUG12) was replaced by $(GU)_3$, $(GU)_6$, or $(GU)_{12}$. A system containing only the protein was also considered to study the conformational changes of TDP-43 upon RNA binding. All MD simulations were carried out using GROMACS software version 2018.2 (*Abraham et al., 2015*) with the "all atom" force field amber ff03 with associated nucleic acid parameters (*Duan et al., 2003*) and periodic boundary conditions. The protonation states of the residues were adjusted to the pH (6.8) used in *NMR* experiments as well as KCl concentration (25 mM) and counter-ions were added to neutralize the system. In all cases, the system was solvated in a box of TIP3P water (*Mahoney and Jorgensen, 2000*). Each system was first energy minimized using 5000 steps of steepest descent, then heated from 0 to 298 K at constant volume for 500 ps and equilibrated in the NPT ensemble at p=1 atm for 500 ps which was followed by 100 ns of NPT production run. The Velocity Rescaling (with $\tau = 0.1$ ps) and Parrinello–Rahman methods were used for temperature and pressure control, respectively (*Parrinello and Rahman, 1981*). The equations of motion were propagated with the leap-frog algorithm and the time step was $\Delta t = 2$ fs. All covalent bond lengths were constrained with LINCS (*Hess, 2008*).

Free energy landscape (FEL) calculations: FEL is represented using two variables, (1) the radius of gyration of the system and (2) the root mean square deviation (RMSD) with respect to the average conformation; reflecting specific properties of the system and measure conformational variability. The Gibbs free energy is estimated from populations (probability distribution) of the system with respect to the previously chosen variables. The 3D representation shows 'valleys' of low free-energy, which represent metastable conformational states of the system, and 'hills', which account for the energetic barriers connecting these states (*Figure 5—figure supplement 4*).

Energy decomposition analysis: An energy decomposition analysis was performed at the dimerization interface to assess potential contributions of local non-bonded interactions to stability. Conformationally averaged energies of the non-bonded energy terms of major contributing residues to enthalpy due to their interaction at the dimerization interface in the RRM1–2/$(GU)_{12}$ complex. Energies were averaged over 100 ns of MD simulation (*Supplementary file 2*).

## AFM imaging and image analysis

A Nanoscope V Multimode 8 (Bruker, Santa Barbara, CA) in PeakForce Tapping (PFT) mode using Scanasyst-Air probes (Bruker) was used to record AFM images in air. Continuous force-distance curves were thus recorded with an amplitude of 100–300 nm at low frequency (1–2 kHz). The point of using the PFT mode is to decrease the lateral and shear forces. Images were recorded at 2048 × 2048 pixels at a line rate of 1.5 Hz.

To adsorb the proteins and DNA on mica, putrescine ($Pu^{2+}$) was added to the solution (20 mM Tris–HCl, pH 7.4 containing 25 mM KCl, 0.5 mM DTT, and 2 mM $MgCl_2$) to a final concentration of 1 mM, after which a 10 µL droplet was deposited on the surface of freshly cleaved mica at room temperature for 30 s and dried for AFM imaging as described previously (*Singatulina et al., 2019*).

## Recombinant protein sedimentation assays

The recombinant full-length TDP-43 proteins were diluted in 40 µL in 20 mM Tris–HCl, pH 7.4 containing 25 mM KCl, 0.5 mM DTT, and 2 mM $MgCl_2$ (Buffer B). After incubation for 5 min of the proteins and oligonucleotide mixtures at 30 °C, samples were centrifuged at 25,000× g for 10 min and the supernatant was transferred to a new tube and SDS–PAGE Sample Loading Buffer was added. The pellets were resuspended using SDS–PAGE Sample Loading Buffer, and the final volume was adjusted by adding 40 µL of Buffer B. The fractions of the pellets and supernatants were analyzed by SDS–PAGE (10 % poly-acrylamide). Gels were stained with Coomassie blue and quantified using an Amersham Typhoon Imagers by scanning with an excitation wavelength of 685 nm.

## Microtubule bench experiments

Plasmids harboring the gene, coding for mutated full-length TDP-43, were obtained by site-directed mutagenesis on the human *TDP-43* gene directly on the Tau-RFP-TDP43 plasmid, a mammalian vector

expressing a TDP-43/RFP/Tau chimera, as previously described (*Maucuer et al., 2018*). The mutagenesis experiments were performed by using the 'Quikchange II XL site-directed mutagenesis kit' and the appropriate oligonucleotides (Eurofins Genomics). The introduced mutations were checked by DNA sequencing. Plasmid encoding Sam68-GFP was prepared as previously described (*Pankivskyi et al., 2021*).

Cell culture experiments: HeLa cells were maintained in DMEM (Dulbecco's modified Eagle's medium) containing 10 % fetal bovine serum (FBS), penicillin and streptomycin (100 µg/mL) (GIBCO Life Technologies). Cells at confluence $10^6$ were plated in four-well plates and co-transfected with the indicated Tau-RFP-TDP43 plasmid bearing point mutations as well as Tau-GFP-TDP43 plasmid harboring the wild-type gene of TDP-43, using lipofectamine 2000 reagent (Invitrogen) as vehicle. Co-transfected cells were incubated during 24 hr at 37 °C in 5 % of $CO_2$. Cells were washed with PBS, then fixed with ice-cold methanol for 10 min at −20 °C, and washed with PBS. Moreover, cells were further fixed with 4 % paraformaldehyde (PFA) diluted in PBS during 30 min at 37 °C. This double methanol/PFA fixation is performed in order to improve the microtubule structure visualization. After final washes with PBS, samples were prepared for fluorescence microscopy imaging.

Image analysis to detect sub-compartmentalization: The image analysis was carried out following the procedure previously detailed (*Maucuer et al., 2018*). Briefly, fluorescence emission was collected with an oil immersed 63×/1.4 NA objective with a Nikon microscope. Fluorescence analysis was performed after processing intensities by filtering out large (shading correction) and small (smoothing) structures (Fast Fourier Transform process) and removing the background value (Subtract background tool, ImageJ). Image analysis was processed with the following parameters: The line thickness used to record changes of fluorescence intensities was 360 nm (three pixels). The length analyzed along the microtubule network was 0.5 mm for each condition. A compartment was detected whenever variation of the RFP/GFP fluorescence ratio exceeds 20 % over a length longer than 720 nm (six pixels). The enrichment of the compartment was obtained by measuring the maximal ratio ($I_{RFP}$-TDP43/$I_{GFP}$-TDP43) or ($I_{GFP}$-TDP43/$I_{RFP}$-TDP43) over the length, L, of the considered compartment. A similar procedure was followed for analyzing GFP-TDP43-rich compartments. Four biological replicates were performed for each condition and values were then processed as previously described (*Maucuer et al., 2018*).

## Small-angle X-ray scattering (SAXS)

X-ray scattering data were collected at the SWING beamline of the SOLEIL Synchrotron (Saint-Aubin, France). All measurements were performed using a Superdex 75 Increase 5/150 GL column (GE Healthcare) on-line with the SAXS measuring cell and a 1.5 mm diameter quartz capillary contained in an evacuated vessel (*David and Pérez, 2009*). All experimental details are given in *Supplementary file 1* in accordance with the guidelines provided by *Trewhella et al., 2017*.

For SAXS experiments, RRM1–2 protein alone and in the presence of $(GU)_3$, $(GU)_6$, or $(GU)_{12}$ at molar ratios reported in *Supplementary file 1* were prepared in SAXS buffer (10 mM Tris–HCl pH 7.6 containing 50 mM KCl and 1 mM TCEP) supplemented with SUPERase·In RNase Inhibitors. In each case, 50 µL of sample was loaded into the column with a flow rate fixed at 0.2 mL/min at 20 °C.

Scattering of the elution buffer before void volume was recorded and used as buffer scattering for subtraction from all protein or protein/RNA patterns.

Data were first analyzed using Foxtrot, a Swing in-house software, and then using the US-SOMO HPLC module (*Brookes et al., 2016*). This program provides for each SAXS frame, the values of both scattering intensity I(0) and radius of gyration $R_g$ by applying the Guinier analysis together with a calculation of the approximate molar mass using the Rambo and Tainer approach (*Rambo and Tainer, 2013*). Identical frames under the main elution peak were selected and averaged for further analysis. All structural parameters (radius of gyration $R_g$ [Å], maximal extension $D_{max}$ [Å], and molar mass M [kDa]) were extracted from averaged SAXS experimental curves (*Supplementary file 1*).

All SAXS intensity calculations were obtained from an ensemble fit conducted on 1000 conformations sampled along the MD trajectory and separated by 100 ps, using GAJOE from the suite EOM (*Tria et al., 2015*; *Bernadó et al., 2007*).

## Stress granule experiments

Stress granules experiments were performed as previously described (*Singatulina et al., 2019*). HeLa cells were cultured in DMEM supplemented with 10 % of FBS in the presence of penicillin

and streptomycin (100 µg/mL) (GIBCO Life Technologies). Cell cultures were maintained at 37 °C in an incubator controlled at 5 % of $CO_2$. Cells were allowed to grow on 12 mm round coverslips. The plasmids were constructed to express, in mammalian cells, the full-length TDP-43 wild type or mutants bearing an HA tag peptide on N-terminal. For transfection experiments, cells were incubated with 0.6 µg of the plasmid by using lipofectamine 2000 reagent (1 µL/sample) as vehicle. Briefly, the complete coding sequence of *Homo sapiens* TDP-43 and the indicated mutants were amplified taking the corresponding plasmids, used in microtubule bench experiments, as template (see above). FspBI/XhoI sites were introduced on the primers which were used for cloning into the pcDNA3HA vector (*Lyabin and Ovchinnikov, 2016*).

Oxidative stress: HeLa cells were treated with puromycin (2.5 µg/ml) 20 min at 37°C and then with hydrogen peroxide ($H_2O_2$) (300 µM) during 30 min at 37 °C in a $CO_2$ controlled chamber. After treatment, cells were washed twice with warm-PBS and fixed with 4 % paraformaldehyde (PFA) diluted in PBS for 30 min at 37 °C. Cells were then incubated with 70 % ethanol during 10 min at room temperature followed by incubation in presence of 1 M Tris–HCl pH 8.0 for 5 min.

In situ hybridization: To visualize mRNA, HeLa cells were incubated with a poly-dT oligonucleotide coupled with Cyn-2 (Molecular Probes Life Tech.) for 2 hr at 37 °C. Washings were carried out using 4× and then 2× SSC buffer (1.75 % NaCl and 0.88 % sodium citrate, pH 7.0). To visualize TDP-43 protein, cells were incubated overnight at 4 °C with an anti-HA primary antibody (Sigma-Aldrich) diluted ($10^{-3}$) in 2 × SSC buffer containing 0.1 % Triton X-100. After washings, cells were incubated with a secondary goat anti-rabbit IgG antibody ($10^{-3}$) coupled to Alexa Fluor Plus 594 (Molecular Probes Life Tech.) for 90 min at room temperature. For nuclei visualization, cells were incubated 30 s with DAPI (0.66 mg/mL) (Sigma-Aldrich).

Image and statistical analysis: mRNA, TDP-43, and nuclei fluorescence emissions were collected at 500, 600, and 460 nm, respectively, with an oil immersed 63×/1.4 NA objective with a Nikon microscope. Exposure times were set up at 700 ms for all emissions wavelengths. Image analysis were processed with a line width of 3 nm (Line tool, ImageJ) used to record fluorescence intensities. Values were then analyzed in order to statistically distinguish the distribution of TDP-43 protein in stressed cells: TDP43-poor stress granules, TDP-43 condensates, and TDP43-rich stress granules containing mRNA (*Maucuer et al., 2018*). Cytoplasm and nuclear enrichment for mRNA-containing TDP-43 stress granules was also considered for both wild-type and mutant forms.

Hela cells were purchased from ATCC (ATCC CCL-2) and were free from mycoplasma contamination. The cell line identity was tested and authenticated (see *Supplementary file 5*).

## Acknowledgements

This work was supported by INSERM, CNRS, University UEVE, Université Paris-Saclay, Genopole, and Région Ile-de-France (SESAME grant n°15013102). This work was also supported by the Association pour la recherche sur la SLA (ARSLA) (eOTP 19SATSAB02 grant to JCRG) and Genopole (Grant for postdoctoral fellowship support to KEH). We acknowledge Synchroton SOLEIL (Saint-Aubin, France) for provision of synchrotron radiation facilities and personal assistance at the SWING beamline.

## Additional information

### Funding

| Funder | Grant reference number | Author |
| --- | --- | --- |
| Genopole | SATURNE 2018-SABNP | Ahmed Bouhss |
| Association pour la recherche sur la SLA (ARSLA) | eOTP 19SATSAB02 | Juan Carlos Rengifo-Gonzalez |
| University Evry UEVE | FRR- Action 2 | Ahmed Bouhss |
| Région Ile-de-France | SESAME grant n°15013102 | Marie-Jeanne Clément |
| Genopole | | Krystel El Hage |

| Funder | Grant reference number | Author |
|---|---|---|

The funders had no role in study design, data collection and interpretation, or the decision to submit the work for publication.

## Author contributions

Juan Carlos Rengifo-Gonzalez, Krystel El Hage, Marie-Jeanne Clément, Emilie Steiner, Dominique Durand, Investigation, Validation; Vandana Joshi, Pierrick Craveur, Investigation; David Pastré, Conceptualization, Formal analysis, Validation, Writing – review and editing; Ahmed Bouhss, Conceptualization, Formal analysis, Investigation, Supervision, Validation, Writing – original draft, Writing – review and editing

## Author ORCIDs

Dominique Durand (ID) http://orcid.org/0000-0001-9414-5857
David Pastré (ID) http://orcid.org/0000-0002-3348-9514
Ahmed Bouhss (ID) http://orcid.org/0000-0002-6492-1429

## Decision letter and Author response

Decision letter https://doi.org/10.7554/eLife.67605.sa1
Author response https://doi.org/10.7554/eLife.67605.sa2

# Additional files

## Supplementary files

• Supplementary file 1. SAXS experimental conditions and deduced parameters from collected data. Characteristic dimensions, Rg and Dmax, and molar mass (MMcorrelation volume) were obtained from data analysis. The theoretical masses (MMsequence) were calculated from the amino acid sequence.

• Supplementary file 2. Bonds involved in the multimerization of TDP-43 as deduced from the complex RRM1–2/(GU)$_{12}$ MD model. Physical parameters of the established interactions between atoms of residues of RRM2 pocket around V220 and those located in RRM1 loop 3 from monomer 1 and 2, respectively, are shown. Values in brackets indicates the energy contribution (in kcal/mol) of amino acid residues to the protein-protein interface stability. Energies were averaged over 100 ns of MD simulation and values are reported in kcal/mol with variant of fluctuations being ±0.1 kcal/mol. [1]bb and vdW correspond to backbone and van der Waals, respectively.

• Supplementary file 3. Poly (GT) repeats binding capacity (N), apparent dissociation constant ($K_D$), and thermodynamic parameters for RRM fragments of TDP-43, as determined by ITC. [a] and [b] correspond to apparent dissociation constants $K_{D1}$ and $K_{D2}$, respectively. The thermodynamic parameters (ΔH, TΔS, ΔG) and $x^2$ values were expressed in kcal/mol and (kcal/mol)$^2$, respectively.

• Supplementary file 4. Poly (GT) repeats binding capacity (N), apparent dissociation constant ($K_D$), and thermodynamic parameters for RRM1–2 protein mutants, as determined by ITC. [a] and [b] correspond to apparent dissociation constants $K_{D1}$ and $K_{D2}$, respectively. The thermodynamic parameters (ΔH, TΔS, ΔG) and $x^2$ values were expressed in kcal/mol and (kcal/mol)$^2$, respectively.

• Supplementary file 5. Cell line authentication.

• Transparent reporting form

## Data availability

All data generated or analysed during this study are included in the manuscript and supporting files.

The following previously published datasets were used:

| Author(s) | Year | Dataset title | Dataset URL | Database and Identifier |
|---|---|---|---|---|
| Lukavsky PJ, Daujotyte D, Tollervey JR, Ule J, Stuani C, Buratti E, Baralle FE, Damberger FF, Allain FHT | 2013 | NMR structure of human TDP-43 tandem RRMs in complex with UG-rich RNA | https://www.rcsb.org/structure/4BS2 | RCSB Protein Data Bank, 4BS2 |

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
