## [Decision Letter]

**Acceptance summary:**

The importance of the interplay between binding and self-association of TDP-43 has been espoused, but never measured or well established. This paper provides an important step in reconstituting the functional vs. dysfunctional network.

**Decision letter after peer review:**

Thank you for submitting your article "The cooperative binding of TDP-43 to GU-rich RNA repeats antagonizes TDP-43 aggregation" for consideration by *eLife*. Your article has been reviewed by 2 peer reviewers, and the evaluation has been overseen by a Reviewing Editor and Cynthia Wolberger as the Senior Editor. The reviewers have opted to remain anonymous.

The Reviewing Editor has drafted this to help you prepare a revised submission.

Essential revisions:

1) Please follow the recommendation of Reviewer 1 and reorganize the manuscript as requested. The proposal is to focus on one salient point per paragraph, a clear description of what is being measured and why, followed by a summary of the findings or a suitable segue to the next paragraph. Similar sentiments were expressed by Reviewer 2 as well.

2) The reviewers found the data regarding cooperative binding to be interest and both reviewers believe that these data will be of interest to a broad audience. However, the approach used to quantify cooperativity poses some challenges for a broad readership. Please consider the investment of space to explain how cooperativity is being quantified, and how these interpretations emerge from the data.

3) Does the impairment of cooperative binding have a direct bearing on the formation of TDP-43 condensates / aggregates? That comes through as one of the main messages, but what was deemed to be missing was a direct illustration of this connection. The question that arises is why is cooperative binding essential for inhibiting aggregation? Does this have to do with the threshold concentrations for cooperative binding vs. that of phase separation? If so, then suitable linkage thermodynamics might provide the appropriate framework. Further, since residues responsible for cooperative binding have been identified, it is recommended that the advice of reviewer 2 be heeded in dissecting how mutations of these residues might impact cooperative binding and hence aggregation.

Addressing these three points in a suitably revised manuscript would be essential for moving forward with the revised version.

*Reviewer #1:*

In this manuscript, Rengifo-Gonzalez, Juan Carlos et al. performed studies to elucidate TDP43-RNA binding interactions and found that tandem TDP43 RNA-recognition motifs (RRMs) bind to GU-repeats in a cooperative manner. Additionally, they utilized NMR spectroscopy structural analysis, in combination with mutagenesis experiments, to identify that this cooperative binding is driven by the interaction between the loop 3 of RRM1 and a pocket centered around the V220 in RRM2. The authors go on to examine the relationship between TDP43-RNA binding and the formation of TDP43-positive inclusions characteristic of ALS/FTLD, and they conclude that disruption of TDP43 cooperative binding promotes the assembly of mRNA-poor TDP43 aggregates.

The authors successfully demonstrate that TDP43 cooperatively binds GU-rich sequences and identify the residues responsible for that interaction, which addresses an outstanding question in the TDP43 field and is likely of interest to potential readers. However, the work could be improved by additional evidence that impaired cooperative TDP43 binding is relevant to the formation of TDP43 aggregates. No distinction is made between cooperative binding to RNA and TDP43 RNA binding in general, and TDP43 overexpression in HeLa cells under two forms of stress (H_2_O_2_ and puromycin) is a highly artificial model of stress granule, not aggregate, formation. It is also not clear what the unique TDP-43 assemblies formed by the mutants are, and how they relate to typically TDP-43 aggregates seen in disease. These studies would benefit from a more relevant model of TDP43 aggregation to identify the relevance and functional consequences of TDP43 cooperative binding on aggregate formation.

While this manuscript contains novel insights on TDP43 cooperative binding that are likely of interest to the field, these findings are not presented clearly. The text is very difficult to follow, primarily because the rationale and conclusions for each section are not clearly stated. We recommend reorganizing the manuscript such that there is one point per paragraph, with a clear explanation of what experiment is being performed and why.

Similarly, the figures concerning cooperative binding are non-intuitive, and we suggest that the biophysical data are presented in a format that is more standard to the field. Additional figure labels throughout the manuscript would further aide in comprehension.

Finally, we suggest a different model system to study the impact of impaired cooperative binding on TDP43 aggregation. Stress granules are transient structures, and there is no definitive evidence that they seed TDP43 aggregates. Overexpression of TDP43 isoforms in immortalized cell lines or a disease-relevant neuronal context (e.g. primary rodent neurons) is sufficient to induce aggregate formation, and would more clearly demonstrate the impact of TDP43 cooperative binding on aggregate formation.

*Reviewer #2:*

In this work, the authors have aimed to precisely map the residues that are involved in TDP-43 cooperative binding to RNA and that may mediate its aggregation. Using an extensive battery in the RRM1 and RRM2 domains, the authors have identified the intermolecular RRM1-2 interface responsible for cooperative binding.

The main strengths of the work are the very extensive structural and molecular characterisation of the various TDP-43-RNA complexes. Moreover, and most importantly, the authors have tested the intermolecular interface using compensatory mutations, thus greatly supporting their conclusions.

There are some unavoidable weaknesses in the work, in that most results are obtained in the absence of the NTD and CTD domains of the protein (this is acknowledged by the authors) and the fact that many important TDP-43 binding substrates are not made up by straight UG-repeats, raising the possibility that cooperative binding to these sequences might differ from the one investigated by the authors in this work.

Nonetheless, the work is likely to be of considerable impact to the field. The novelty is that it starts to address at the molecular level one of the most disease related properties of TDP-43 protein represented by cooperative binding.

There are a several areas of improvements that could be considered by the authors:

a) first of all, and most importantly, it would have been very interesting to see what happens to the cooperativity in the presence of substrates other than UG-repeats. For example, the sequence known as CLIP34nt (A Bhardwaj et al., 2013) plays an important role in TDP-43 auto regulation. Would the cooperativity of TDP-43 on this substrate differentiations noticeably from UG24-repeats?

b) Disease-associated mutations are few in the RRM1 and RRM2 regions but some have been described (ie. K181E). Could some of these be considered as affecting the cooperative binding process of TDP-43. This should be discussed.

Finally, the rather less detailed part of the manuscript is represented by the nuclear and cytoplasmic condensates of the mutant TDP-43s in cells. Although quite interesting, these condensates could have been characterised better with just small additional effort. For example:

c) The TDP-43 condensates observed in the nucleus under oxidative stress conditions co-localize with some specific nuclear structures such as nuclear stress bodies or other well-known nuclear bodies?.

d) In Figure 7b, it is not clear how the cytoplasmic TDP-43 condensates do not relate to stress granules. In the manuscript it is stated that they do not co-localize with mRNA which would suggest this to be the case. However, have the authors performed some staining with stress granule markers (G3BP, TIA-1) to confirm that TDP-43 does not co-localize (or not) with these structures?

e) Finally, it would have been interesting to know whether these TDP-43 condensates triggered by the key mutations (K140A/T141A, T141A/G142A, G146A, and Q213A) were also positive for phosphorylation at the 409/410 Ver residues.

---

## [Author Response]

Essential revisions:1) Please follow the recommendation of Reviewer 1 and reorganize the manuscript as requested. The proposal is to focus on one salient point per paragraph, a clear description of what is being measured and why, followed by a summary of the findings or a suitable segue to the next paragraph. Similar sentiments were expressed by Reviewer 2 as well.

The manuscript has been rewritten as proposed by the reviewers.

2) The reviewers found the data regarding cooperative binding to be interest and both reviewers believe that these data will be of interest to a broad audience. However, the approach used to quantify cooperativity poses some challenges for a broad readership. Please consider the investment of space to explain how cooperativity is being quantified, and how these interpretations emerge from the data.

In the revised manuscript, we quantified the cooperative binding of TDP-43 by using the ratio of two Kds defining the cooperativity constant as describe by Brown (Int J Mol Sci, 2009). The dissociation constant Kd1 accounts for the affinity of the first RRM1-2 monomer that binds to GU repeats and Kd2 accounts for the affinity of the second binding event. A ratio kd1/kd2 >1 means that the second binding event has a higher affinity than the first one (positive cooperativity). A table recapitulating the cooperativity score has been added to Figure 1. It should be easier to understand by a broad audience.

3) Does the impairment of cooperative binding have a direct bearing on the formation of TDP-43 condensates / aggregates? That comes through as one of the main messages, but what was deemed to be missing was a direct illustration of this connection. The question that arises is why is cooperative binding essential for inhibiting aggregation? Does this have to do with the threshold concentrations for cooperative binding vs. that of phase separation? If so, then suitable linkage thermodynamics might provide the appropriate framework.

The manuscript has been rearranged to clearly enlighten how our data on the cooperative binding of TDP-43 to mRNA may matter to keep TDP-43 soluble. Notably, we show that the presence of GT repeats preserves TDP-43 solubility in vitro but to a lesser extent for the G146A mutant (Figure 6a and b). The linkage associating the thermodynamics and the role of cooperative binding of TDP-43 to RNA in liquid-liquid phase separation is an interesting question that deserves, in our view, a study on its own with a theoretical and experimental approach. We prefer here to keep a simple message: the observed increase in TDP-43 solubility is due to the separation distance between self-adhesive N-terminal domains and C-terminal domains when TDP-43 proteins are cooperatively associated along GU-rich tracks. We tried to better highlight this point in Figure 9a.

Further, since residues responsible for cooperative binding have been identified, it is recommended that the advice of reviewer 2 be heeded in dissecting how mutations of these residues might impact cooperative binding and hence aggregation.

We have followed the reviewer 2 advice. So far, pathological TDP-43 mutations in ALS patients such as K181E do not correspond to the critical residues for TDP-43 cooperative binding to RNA. In our view, cooperativity-deficient mutants induced a marked phenotype following TDP-43 expression in HeLa cells. Most probably, mutations that would occur naturally would be lethal or would not allow the brain development in adults. However, although K181E was not selected in our screening in HeLa cells, it is indeed possible that mutation such as K181E may slightly impact TDP-43 cooperativity which may be sufficient to promote TDP-43 aggregation in adult neurons in ALS patients. It remains however to be demonstrated.

Addressing these three points in a suitably revised manuscript would be essential for moving forward with the revised version.Reviewer #1:[…] The authors successfully demonstrate that TDP43 cooperatively binds GU-rich sequences and identify the residues responsible for that interaction, which addresses an outstanding question in the TDP43 field and is likely of interest to potential readers. However, the work could be improved by additional evidence that impaired cooperative TDP43 binding is relevant to the formation of TDP43 aggregates. No distinction is made between cooperative binding to RNA and TDP43 RNA binding in general, and TDP43 overexpression in HeLa cells under two forms of stress (H_2_O_2_ and puromycin) is a highly artificial model of stress granule, not aggregate, formation. It is also not clear what the unique TDP-43 assemblies formed by the mutants are, and how they relate to typically TDP-43 aggregates seen in disease. These studies would benefit from a more relevant model of TDP43 aggregation to identify the relevance and functional consequences of TDP43 cooperative binding on aggregate formation.While this manuscript contains novel insights on TDP43 cooperative binding that are likely of interest to the field, these findings are not presented clearly. The text is very difficult to follow, primarily because the rationale and conclusions for each section are not clearly stated. We recommend reorganizing the manuscript such that there is one point per paragraph, with a clear explanation of what experiment is being performed and why.

In the revised manuscript, we present rationale and conclusions for each section to make the article easier to read and follow.

Similarly, the figures concerning cooperative binding are non-intuitive, and we suggest that the biophysical data are presented in a format that is more standard to the field. Additional figure labels throughout the manuscript would further aide in comprehension.

We provide the kd_1_/kd_2_ ratio to estimate the cooperative binding of TDP-43 to mRNA. We also further improved the comprehension of the figures by using a table added to figures 1 and 4. Kd_1_ and kd_2_ are the dissociation constants accounting for the affinities of the first and second TDP-43 proteins that bind to GU repeats. A cooperativity should induce a higher affinity of the second than the first protein that binds to GU repeats (kd_2_<kd_1_).

Finally, we suggest a different model system to study the impact of impaired cooperative binding on TDP43 aggregation. Stress granules are transient structures, and there is no definitive evidence that they seed TDP43 aggregates. Overexpression of TDP43 isoforms in immortalized cell lines or a disease-relevant neuronal context (e.g. primary rodent neurons) is sufficient to induce aggregate formation, and would more clearly demonstrate the impact of TDP43 cooperative binding on aggregate formation.

We agree with the reviewer. As indicated in the manuscript, stress granules may have a positive or neutral impact on TDP-43 aggregation in the cytoplasm. Though this is a very interesting subject of research, the presence of stress granules in neurons of ALS patients is not yet certain. Here we used cellular stress because we obtained a robust aggregation in cells expressing TDP-43 mutants. However, as suggested by reviewer 2, we also performed an analysis of TDP-3 aggregation in the cytoplasm after the expression of TDP-43 mutants without additional stress. Though most cells display a homogenous distribution of TDP-43 in the cytoplasm and nucleus (Figure 7—figure supplement 18), we measured a significant aggregation of TDP-43 mutants. In contrast with wild type TDP-43, aggregates of TDP-43 mutants are distinct from stress granules and are resistant to cycloheximide pretreatment. These new data are shown in the revised manuscript (Figure 7, a new figure).

Reviewer #2:[…] There are a several areas of improvements that could be considered by the authors:a) first of all, and most importantly, it would have been very interesting to see what happens to the cooperativity in the presence of substrates other than UG-repeats. For example, the sequence known as CLIP34nt (A Bhardwaj et al., 2013) plays an important role in TDP-43 auto regulation. Would the cooperativity of TDP-43 on this substrate differentiations noticeably from UG24-repeats?

We performed new ITC experiments to analyze the cooperative binding of TDP-43 RRM1-2 to CLIP34nt (DNA), as suggested by the reviewer. In the case of CLIP34nt, we observed three distinct plateaus supporting the binding of three protein units. Due to the number of binding events (at least three) during titration, the fitting of the experimental data becomes more challenging (Author response image 1) . Data fitting, using a one set of sites model (Malvern), reveals that RRM1-2 displays a significantly lower affinity for CLIP34nt than GT repeats with an apparent Kd of 270 nM (Bhardwaj et al. (ref 64) have reported a value of 112 nM by using EMSA). The lower affinity of TDP-43 for CLIP34nt compared to (GT)12 repeats (more than 2 orders of magnitude) can be explained by a strong preference of this protein for GT or GU base repeats which are little represented in CLIP34nt.

**Author response image 1. sa2fig1:** Binding of RRM1-2 to CLIP24nt oligonucleotide. The binding profile was monitored by ITC and the titration curve was fitted (bottom). The plateaus observed for each oligonucleotide are indicated in red.

We indicate in the discussion of the revised manuscript that exploring the role of nucleic acid sequence on the cooperative behavior of TDP-43 is of interest to understand the formation of TDP-43 clusters on specific mRNA sequences.

b) Disease-associated mutations are few in the RRM1 and RRM2 regions but some have been described (ie. K181E). Could some of these be considered as affecting the cooperative binding process of TDP-43. This should be discussed.

In our extensive screening, we did not identify any pathological mutation as critical for the cooperative association of TDP-43 to RNA. Given the strong phenotype observed in vitro cultured cells with cooperativity-defective mutants such as G146A, we may expect, in a functional point of view, that these mutations would prevent development of an adult brain in animals.

Consequently, while we have no evidence that, for example, K181E would impact the cooperative binding to RNA, we cannot exclude that K181E may have a very slight impact on the cooperative binding which in turn would contribute to the onset of ALS. We need to increase the sensitivity of our detection scheme to find whether other residues may not slightly impact TDP-43 cooperativity through an allosteric mechanism for instance. A sentence was added in the discussion regarding this point.

Finally, the rather less detailed part of the manuscript is represented by the nuclear and cytoplasmic condensates of the mutant TDP-43s in cells. Although quite interesting, these condensates could have been characterised better with just small additional effort. For example:c) The TDP-43 condensates observed in the nucleus under oxidative stress conditions co-localize with some specific nuclear structures such as nuclear stress bodies or other well-known nuclear bodies?.

We performed an additional experiment to determine whether Sam68, an mRNA-binding protein found in nuclear stress bodies is recruited in G146A nuclear aggregates. SAM68 was not detected in G146 nuclear inclusions.

d) In Figure 7b, it is not clear how the cytoplasmic TDP-43 condensates do not relate to stress granules. In the manuscript it is stated that they do not co-localize with mRNA which would suggest this to be the case. However, have the authors performed some staining with stress granule markers (G3BP, TIA-1) to confirm that TDP-43 does not co-localize (or not) with these structures?

We agree that is important to characterize these aggregates. We observed that nuclear and cytoplasmic aggregates of TDP-43 mutants are not enriched in mRNA (poly-dT probes), in FUS (a nuclear RNA-binding protein that also binds to introns). In addition, we shown that wild type TDP-43 co-localized with G3BP-1 positive stress granules (mRNA-rich and sensitive to cycloheximide). In contrast, TDP-43 cytoplasmic aggregates are indeed not positive to anti-G3BP-1 and anti-FMRP antibodies, not enriched in mRNA and unsensitive to cycloheximide.

e) Finally, it would have been interesting to know whether these TDP-43 condensates triggered by the key mutations (K140A/T141A, T141A/G142A, G146A, and Q213A) were also positive for phosphorylation at the 409/410 Ver residues.

We performed these novel experiments with a commercial anti-phospho TDP-43 antibody (409/410 Ser, see Author response image 2, Chen and Cohen, 2019, doi: 10.1074/jbc.RA118.006351; Cykowski et al., 2018, doi: 10.1186/s40478-018-0528-y). We found a significant staining of TDP-43 rich stress granules when wild type TDP-43 is expressed. In addition, G146A and T141A/G142A cytoplasmic or nuclear aggregates are also positive to this antibody.

**Author response image 2. sa2fig2:** Images of HeLa cells expressing HA-tagged wild Type TDP-43 or G146A. As indicated by the red arrow, we detected the presence of TDP-43 in stress granules that appear to be positive for phosphorylation. When G146A is expressed, both cytoplasmic and nuclear aggregates are also positive for phosphorylation (red arrows). Similar results were obtained with K140A/T141A. We also noted the presence of bright anti-phospho TDP-43 spots that are negative to G146A or wild type TDP-43 in some cells, for unknown reasons.

However, we considered that the link between TDP-43 phosphorylation and the recruitment of TDP43 in stress granules or the formation cytoplasmic inclusions deserves a detailed study on its own. We therefore preferred to add a sentence about the relevance of studying TDP-43 phosphorylation in the discussion.